 

# Diverse roles of TssA-like proteins in the assembly of bacterial type VI secretion systems

Johannes Paul Schneider[1] iD, Sergey Nazarov[1,†] iD, Ricardo Adaixo[2], Martina Liuzzo[1], Peter David Ringel[1,‡], Henning Stahlberg[2] iD & Marek Basler[1,]* iD

## Abstract

Protein translocation by the bacterial type VI secretion system (T6SS) is driven by a rapid contraction of a sheath assembled around a tube with associated effectors. Here, we show that TssA-like or TagA-like proteins with a conserved N-terminal domain and varying C-terminal domains can be grouped into at least three distinct classes based on their role in sheath assembly. The proteins of the first class increase speed and frequency of sheath assembly and form a stable dodecamer at the distal end of a polymerizing sheath. The proteins of the second class localize to the cell membrane and block sheath polymerization upon extension across the cell. This prevents excessive sheath polymerization and bending, which may result in sheath destabilization and detachment from its membrane anchor and thus result in failed secretion. The third class of these proteins localizes to the baseplate and is required for initiation of sheath assembly. Our work shows that while various proteins share a conserved N-terminal domain, their roles in T6SS biogenesis are fundamentally different.

**Keywords** ImpA_N domain proteins; protein localization; protein–protein interactions; sheath assembly; type VI secretion system
**Subject Categories** Microbiology, Virology & Host Pathogen Interaction
**The EMBO Journal (2019) 38: e100825**

## Introduction

The type VI secretion system (T6SS) is a membrane-anchored contractile nanomachine used by many Gram-negative bacteria to deliver proteins from the cytosol directly to an extracellular space or across a target cell membrane. The nanomachine structurally and functionally resembles the contractile tail of bacteriophages and R-type pyocins (Brackmann *et al*, 2017). T6SS biogenesis proceeds in a strictly hierarchical order (Wang *et al*, 2019). First, a protein complex spanning both membranes forms (Aschtgen *et al*, 2010; Felisberto-Rodrigues *et al*, 2011; Durand *et al*, 2015). On this

membrane complex, the baseplate forms and subsequently the sheath-tube complex polymerizes (Brunet *et al*, 2015; Nguyen *et al*, 2017; Wang *et al*, 2017; Nazarov *et al*, 2018). Once fully extended, the sheath rapidly contracts, propelling the inner tube and associated effector proteins out of the cell (Basler *et al*, 2012; Cianfanelli *et al*, 2016). The contracted sheath is then disassembled by an ATP-dependent unfoldase, and the components are recycled for another round of sheath extension and contraction (Bönemann *et al*, 2009; Basler & Mekalanos, 2012; Kapitein *et al*, 2013).

Based on phylogenetic analyses, T6SSs have been classified into three different types (i, ii, and iii) and type i can be further split into subclasses i1, i2, i3, i4a, i4b, and i5 (Barret *et al*, 2013; Russell *et al*, 2014; Li *et al*, 2015). The current model of T6SS[i] includes a minimal set of 13 proteins to assemble a functional T6SS (Mougous *et al*, 2006; Pukatzki *et al*, 2006). TssJ, TssL, and TssM form the membrane complex (Brunet *et al*, 2015; Durand *et al*, 2015); VgrG, TssE, TssF, TssG, and TssK form the baseplate (Nguyen *et al*, 2017; Nazarov *et al*, 2018); and TssB/VipA, TssC/VipB, and Hcp form the long sheath-tube polymer (Wang *et al*, 2017; Szwedziak & Pilhofer, 2019). The AAA(+) ATPase ClpV disassembles contracted sheath, and its subunits can be reused to build another T6SS sheath (Bönemann *et al*, 2009; Basler & Mekalanos, 2012; Basler *et al*, 2012; Kapitein *et al*, 2013; Förster *et al*, 2014; Douzi *et al*, 2016). T6SS[ii] is exclusively populated by the *Francisella* pathogenicity island and contains a set of 17 core components (Bröms *et al*, 2010; de Bruin *et al*, 2011), while the type iii system is found only in *Bacteroidetes* and contains 12 core components (De Maayer *et al*, 2011; Russell *et al*, 2014). Recently, a fourth type (T6SS[iv]) that is closely related to extracellular injection machineries such as R-type pyocins and antifeeding prophages has been described (Böck *et al*, 2017). This particular system does not contain a canonical trans-membrane anchor and also lacks ClpV unfoldase (Böck *et al*, 2017). However, similarly to *Francisella*, contracted sheaths may be refolded by a related ATPase (Brodmann *et al*, 2017).

TssA proteins have been initially shown to play an essential role in assembly of baseplate and the sheath-tube in *Pseudomonas aeruginosa* (TssA1[PA]) and *Escherichia coli* (TssA[EC]) (Planamente *et al*, 2016; Zoued *et al*, 2016). Interestingly, TssAs can be categorized into different classes harboring distinct protein domain

1  Biozentrum, University of Basel, Basel, Switzerland
2  Center for Cellular Imaging and NanoAnalytics (C-CINA), Biozentrum, University of Basel, Basel, Switzerland
   *Corresponding author. Tel: +41 61 207 21 10; E-mail: marek.basler@unibas.ch
   †Present address: Interdisciplinary Center for Electron Microscopy (CIME), EPFL, Lausanne, Switzerland
   ‡Present address: Institute of Forensic Medicine, Justus-Liebig-University Giessen, Giessen, Germany

architectures. The specific architecture presumably affects their function during biogenesis of the T6SS. All TssA-like proteins harbor a conserved ImpA_N domain (PF06812) located at the N-terminal end, while the C-terminal part differs in its composition. Recent analyses showed that domains can be further segregated into ImpA containing domain (Nt1), middle domain (Nt2), and C-terminal domain (CTD) (Dix *et al*, 2018). A high-resolution crystal structure of $TssA_{EC}$ C-terminus harboring a VasJ domain (PF16989) was recently obtained, and it was shown that this part of the protein forms two stacked hexameric rings (Zoued *et al*, 2016). Dynamic rearrangement of wedges connecting the six helices supposedly leads to a ~90 Å opening of the structure. $TssA_{EC}$ interacts with membrane complex and baseplate components, but also with sheath component TssC/VipB. Thus, it was proposed that $TssA_{EC}$ might coordinate sheath-tube assembly and guarantee its stability in the extended state (Zoued *et al*, 2016). However, another recent study showed that the C-terminal domain (CTD, G388-L472, helices α8–α12) of a closely related TssA from *Aeromonas hydrophila* forms a high-order oligomer with D5 symmetry (Dix *et al*, 2018). The CTD is connected to the middle N-terminal domain (Nt2, R232-L374, helices α1–α7) through a ~21 residues flexible linker. Neighboring Nt2 domains form dimers, which do not follow D5 symmetry of the CTD oligomer (Dix *et al*, 2018).

$TssA1_{PA}$ was suggested to contain partial secondary structure homologies to the phage baseplate component gp6 in its C-terminal part (Planamente *et al*, 2016). It forms a dodecameric ring with dimensions that are similar to sheath-tube ring and has a central hole that could accommodate Hcp. $TssA1_{PA}$ was further shown to interact with baseplate components TssK, TssF, and VgrG1a, sheath-tube and ClpV, but in contrast to $TssA_{EC}$ not with components of the membrane complex. Due to these properties, it was proposed that $TssA1_{PA}$ might be a baseplate component (Planamente *et al*, 2016).

Lastly, some TssA-like proteins harbor a transmembrane region and a C-terminal VasL domain of unknown function (PF12486). These TssAs were suggested to play an accessory role and corresponding genes thus referred to as *tagA* (type VI secretion accessory gene with ImpA domain) (Zoued *et al*, 2017). Recently, a TagA protein from *E. coli* ($TagA_{EC}$) was shown to interact with $TssA_{EC}$, localize at the distal end of sheath once it was fully extended and to stabilize the extended structure. Deletion of $tagA_{EC}$ caused excessive sheath polymerization, bending, and breaking of sheath structures and thus reduced efficiency of killing target cells (Santin *et al*, 2018). In addition, $TagA_{EC}$ was shown to be required for contraction of a part or the full-length sheath toward the distal end. While it is unclear if these non-canonical sheath contraction events result in protein secretion, these contractions constitute up to one-third of observed contractions in *E. coli* (Szwedziak & Pilhofer, 2019).

Importantly, certain TssA proteins seem to be indispensable for proper T6SS assembly. Hcp secretion was not detectable in a $\Delta tssA_{EC}$ strain, and no sheath structures were observed in a $\Delta tssA_{EC}$ TssB-mCherry strain (Zoued *et al*, 2016). Similarly, secretion of Hcp, VgrG1a, and Tse3 was not detectable in a $tssA1_{PA}$ knockout strain and sheath formation in a TssB1-sfGFP $\Delta tssA1_{PA}$ strain was severely decreased (Planamente *et al*, 2016).

Here, we investigated the role of several distinct proteins sharing the ImpA_N domain and we show that their functions differ

significantly. The *Vibrio cholerae* and *P. aeruginosa* TssA proteins $TssA_{VC}$ and $TssA2_{PA}$ with C-terminal VasJ domain (Class A) facilitate sheath assembly initiation and polymerization by forming a stable dodecamer at the end of the sheath that is distal from the membrane anchor. The TagA protein of *V. cholerae* ($TagA_{VC}$) with the ImpA_N domain followed by a hydrophobic domain (Class B) localizes to cell membrane and prevents sheath assembly likely by competing with $TssA_{VC}$. Finally, we show that a third class of ImpA_N domain containing proteins (Class C), represented by TssA1 in *P. aeruginosa*, localizes to the site of sheath assembly initiation. Our data show that ImpA_N domain containing proteins have diverse functions in the biogenesis of T6SS and that their role in sheath assembly is likely dictated by the structure and function of their C-terminal domains.

## Results

### $TssA_{VC}$ and $TssA2_{PA}$ facilitate sheath assembly initiation and polymerization

Proteins that have ImpA_N domain followed by C-terminal VasJ domain form Class A of TssA-like proteins (Fig EV1A). This class is represented by *E. coli* $TssA_{EC}$ (EC042_4540) and *A. hydrophila* $TssA_{AH}$ (AHA1844) (Zoued *et al*, 2016; Dix *et al*, 2018), as well as *V. cholerae* $TssA_{VC}$ (VCA0119). $TssA_{VC}$ and $TssA_{EC}$ share 19.9% sequence identity, while $TssA_{VC}$ and $TssA_{AH}$ share 32.8% sequence identity (Appendix Table S1). To investigate the role of these proteins, we first imaged VipA-mCherry2 sheath assembly in *V. cholerae* in the presence or absence of $tssA_{VC}$ (Appendix Table S2). We found that the parental strain usually forms five T6SS structures at any given time during logarithmic growth (Fig 1A, Movie EV1). This is in agreement with number of sheaths detected in the strain expressing VipA-msfGFP (Vettiger & Basler, 2016). Image analysis showed that the $tssA_{VC}$-negative strain formed mostly dynamic sheath spots and only few structures that fully extended and contracted (Fig 1A, Movie EV1). In a bacterial competition assay, $tssA_{VC}$ knockout strain was able to kill *E. coli* prey cells only at a reduced rate (Fig EV1B). This resembled the reduced prey cell killing by a strain lacking the baseplate component *tssE*, which forms about 1000 times less structures compared to the parental strain (Vettiger & Basler, 2016). In addition, speed of sheath polymerization dropped from 23 nm per second measured in the parental strain to 3 nm per second in the strain lacking $tssA_{VC}$ (Fig EV1C).

Similarly, we analyzed a second member of Class A, the TssA2 from H2-T6SS of *P. aeruginosa* (PA1656) (Sana *et al*, 2012; Allsopp *et al*, 2017). $TssA2_{PA}$ shares 21,8% sequence identity with $TssA_{VC}$ and 25.2% sequence identity with $TssA_{EC}$ (Appendix Table S1). We show that multiple structures of TssB2-mCherry2 sheath reside in single cells (Fig 1B, Movie EV2). Dynamics of H2-T6SS sheaths are, however, significantly slower than *V. cholerae* sheaths. Full extension of one sheath can take up to 10 min, and sheath structures stay in extended state for at least 5 min (Fig 1B). Deletion of $tssA2_{PA}$ severely decreased number of T6SS sheaths and mostly dynamic spots were visible; however, few fully extending and contracting sheath structures could be observed (Figs 1B and EV1D, Movie EV2).

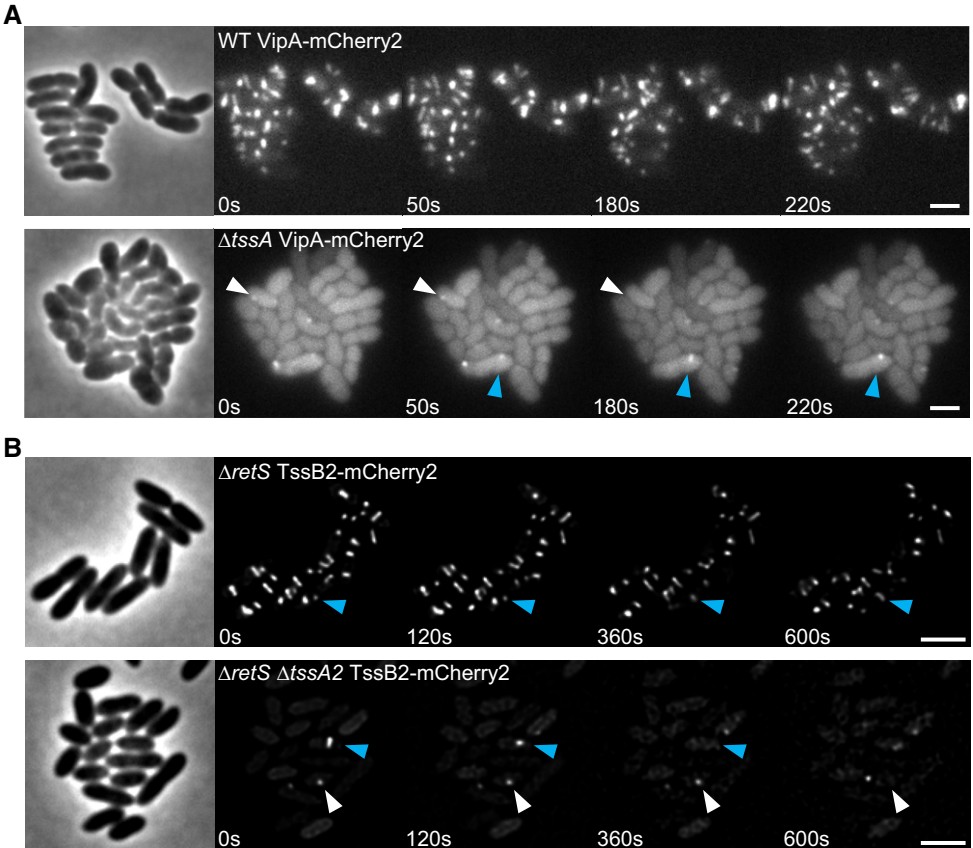

**Figure 1.  TssA$_{VC}$ and TssA2$_{PA}$ influence T6SS sheath assembly dynamics.**

A  Time lapse images of T6SS activity in tagged parental strain (VipA-mCherry2) and Δ*tssA* mutant background. The parental strain forms multiple dynamic structures per cell. Δ*tssA* mutant predominantly forms dynamic sheath spots (white arrow) and few WT-like sheath structures (blue arrow). Scale bars: 2 μm.

B  H2-T6SS dynamics in tagged parental strain (Δ*retS* TssB2-mCherry2, referred to as WT H2) and Δ*tssA2$_{PA}$* strain. Blue arrow indicates extending and contracting T6SS sheath structures, white arrow points to dynamic sheath spots. The WT H2 strain harbors multiple dynamic sheath structures per cell, while the Δ*tssA2$_{PA}$* strain displays very few dynamic spots or extended and contracting structures. Scale bars: 2 μm.

## TssA$_{VC}$ and TssA2$_{PA}$ localize to the distal end of an assembling sheath

Previous work of Zoued *et al* demonstrated that another member of TssA Class A, TssA$_{EC}$ of *E. coli*, first localizes to membrane complex and then coordinates sheath-tube assembly at the distal end, presumably by incorporating new tube and sheath components (Zoued *et al*, 2016). Since TssA of *V. cholerae* is closely related to TssA$_{EC}$ (Appendix Table S1), we wondered if the two proteins share similar role in T6SS biogenesis. We fused mNeonGreen to TssA$_{VC}$ and observed its localization using fluorescence microscopy in a strain background with mCherry2-tagged sheath (VipA-mCherry2). While sheath assembly was about two times slower (Fig EV1C), the T6SS in TssA$_{VC}$-mNeonGreen/VipA-mCherry2 remained fully functional (Fig EV1B). We found that, in most cases, TssA$_{VC}$ first localized to T6SS assembly initiation site before sheath signal appeared and then colocalized with a distal end of a polymerizing sheath (Fig 2A, Movie EV3).

For analysis of sheath dynamics in time lapse movies, we generally used kymographs generated with Fiji (Schindelin *et al*, 2012). A straight line was drawn along an assembling sheath, and the signals

of the underlying pixels were replotted in a new XY coordinate system where the pixels along the *Y*-axis represent the pixels along the line drawn over an assembling sheath and the individual time points are shown along the *X*-axis. Such representation allows simple visualization of sheath assembly, measurement of sheath length, and speed of assembly as well as detection of colocalization of two proteins. Kymographs of TssA$_{VC}$-mNeonGreen and VipA-mCherry2 movies of 300 cells revealed that in about two-thirds of the analyzed cases TssA$_{VC}$ stayed attached to assembling sheaths until their contraction shortly (< 5 s) after full assembly (Fig 2B). In about 27% of the cases, TssA$_{VC}$ dissociated from the sheath after its extension to the opposite side of the cell, which was followed by prolonged period of stable extended sheath (Fig 2C). However, in few cases (about 3%) TssA also dissociated during sheath assembly causing a delay or stalling of the sheath polymerization while TssA re-association resumed polymerization (Fig 2D). In about 3% of the cases, TssA$_{VC}$ stayed attached to the sheath distal end even after its contraction (Fig 2E).

To test whether TssA forms a stable complex on the sheath end, we photobleached cytosolic TssA$_{VC}$-mNeonGreen subunits during sheath polymerization. This was achieved by incubating the cells in

the presence of ampicillin, which leads to formation of large viable spheroplasts with functional T6SS (Vettiger et al, 2017). Large cells allow controlled photobleaching of a relatively small section of the cell. We found that $TssA_{VC}$-mNeonGreen signal of the complex at

the distal end of the polymerizing sheath remained constant after photobleaching of the cytosol, suggesting that the $TssA_{VC}$ subunits are not exchanged with the cytosolic $TssA_{VC}$ pool during sheath polymerization (Appendix Fig S1, Movie EV4).

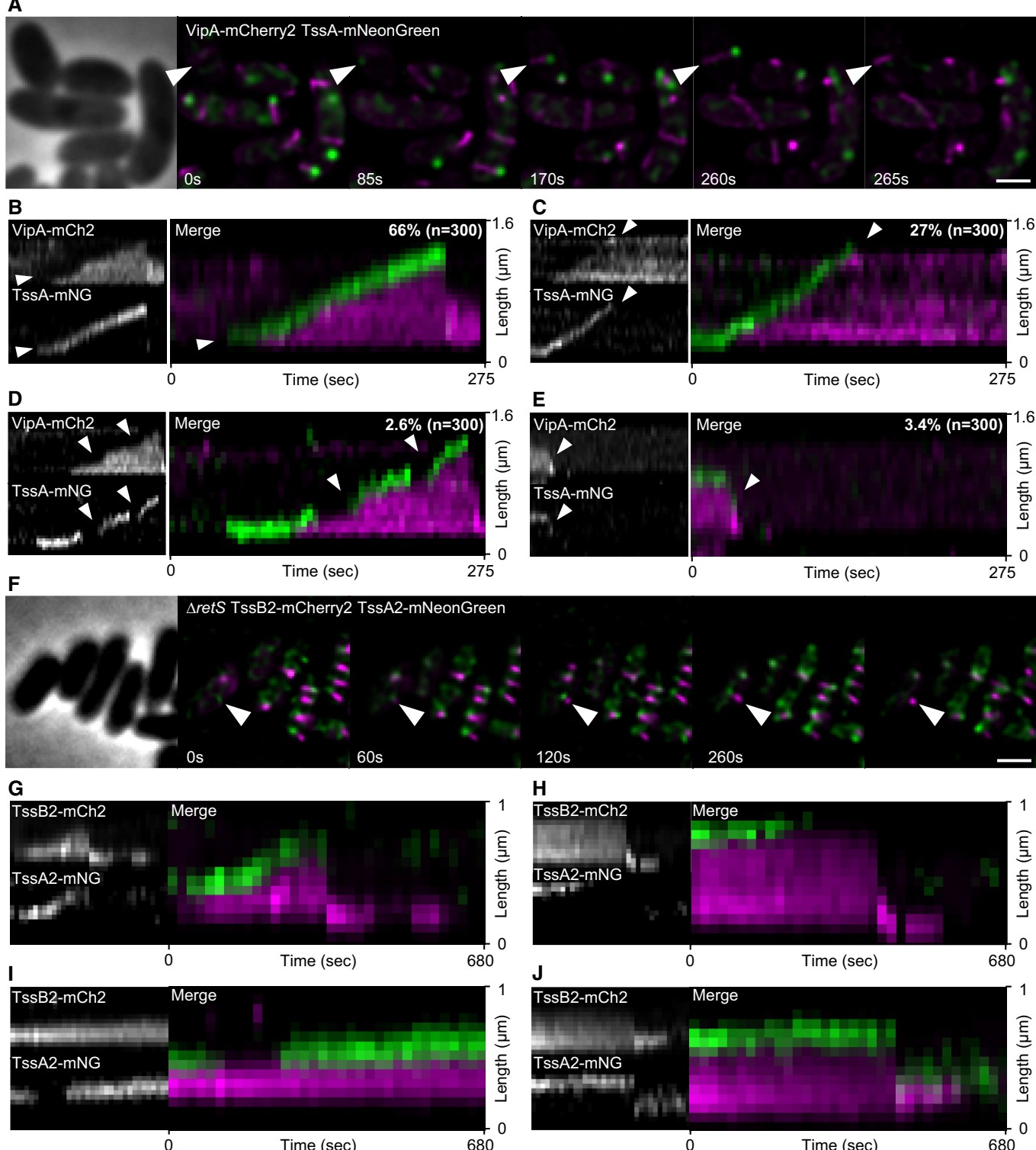

Figure 2.

**Figure 2.  Localization and dynamics of TssA$_{VC}$ and TssA2$_{PA}$.**

A    Time lapse images of T6SS activity in tagged strain (VipA-mCherry2 TssA$_{VC}$-mNeonGreen). White arrow indicates path of TssA$_{VC}$ localization during one cycle of T6SS dynamics. Scale bar: 1 μm.

B–E  Kymographs of dynamics observed for TssA$_{VC}$. Frequency of specific dynamics is indicated in the upper right corner of the merge image. (B) Sheath and TssA$_{VC}$ dynamics from A. TssA$_{VC}$ localizes to T6SS initiation site and to distal end of growing sheath (white arrow). (C) Once sheath reaches cell periphery, TssA$_{VC}$ dissociates from distal end (white arrow) and sheath stays extended a prolonged period of time. (D) TssA$_{VC}$ dissociates from growing sheath structure and re-associates at a later time point (white arrows) to continue catalysis of polymerization. (E) TssA$_{VC}$ stays attached to sheath distal end after contraction (white arrow).

F    Fluorescence microscopy of T6SS dynamics in a double-tagged strain (Δ*retS* TssB2-mCherry2 TssA2$_{PA}$-mNeonGreen). White arrow indicates full cycle of extending and contracting T6SS sheath. Scale bar: 1 μm.

G–J  Kymographs of TssB2 and TssA2$_{PA}$ dynamics. TssA2$_{PA}$ localizes at sheath initiation site and on distal end during polymerization (G), dissociates before contraction (H), dissociates and re-associated during sheath extension (I), or stays attached to distal end after contraction (J).

Since TssA2$_{PA}$ is in the same TssA class as TssA$_{VC}$ and TssA$_{EC}$ (Class A), we hypothesized that all TssA proteins from this class might play the same role in T6SS biogenesis. Consequently, we used fluorescence microscopy to observe TssA2$_{PA}$-mNeonGreen dynamics in TssB2-mCherry2 strain (Fig 2F–J, Movie EV5). We found that TssA2$_{PA}$ displays almost identical dynamics to TssA$_{VC}$. TssA2$_{PA}$ localized to T6SS assembly initiation sites before sheath polymerization and then localized to the distal end of the polymerizing sheath (Fig 2G). Further, we often detected TssA2$_{PA}$ dissociating from sheath distal end after full extension (Fig 2H) or during extension followed by its re-association to the polymerizing sheath (Fig 2I), but also residing on contracted sheath structures (Fig 2J). This suggests that all TssAs within the Class A play the same role in T6SS assembly.

## TssA$_{VC}$ forms a dodecamer *in vivo*

Recent *in vitro* studies have shown that TssAs from *E. coli*, *A. hydrophila,* and *Burkholderia cenocepacia* form rings with 6-fold, 5-fold, and 16-fold symmetry (Zoued *et al*, 2016; Dix *et al*, 2018). We aimed to estimate the oligomeric state of the TssA$_{VC}$ complex *in vivo*. We used the LacI–*lacO* system where two LacI repressor molecules, lacking tetramerization domain, bind to one *lacO* operator sequence (Belmont & Straight, 1998; Dong *et al*, 1999). First, we generated strains harboring 3, 6, or 12 copies of the *lacO* integrated into *V. cholerae* chromosome at the site of the disrupted *lacZ* gene (Fig EV2A). Expressing the *lacI* repressor fused to mNeonGreen (LacI-mNeonGreen) in these strains yielded fluorescent LacI-mNeon-Green spots with 6, 12, or 24 copies of mNeonGreen, respectively (Fig EV2B). The expression of LacI-mNeonGreen in the parental strain lacking *lacO* sequences yielded no detectable foci (Fig EV2B). Similarly, no foci were detected when LacI-mNeonGreen was expressed in

a strain harboring 12 copies of *lacO* but supplemented with IPTG to disrupt binding of LacI to *lacO*. We quantified the signal emitted from LacI-mNeonGreen spots and TssA$_{VC}$-mNeonGreen signal using ImageJ (Fig EV2C and D). Signals observed for TssA$_{VC}$-mNeonGreen fusions were most similar to the signal produced by LacI-mNeon-Green molecules binding to 6 copies of *lacO*. This suggests that TssA$_{VC}$ forms a dodecamer *in vivo*.

## High-resolution structure of TssA$_{VC}$

To further analyze the TssA of *V. cholerae*, we purified the protein and used cryo-electron microscopy (cryo-EM) to solve its structure (Fig 3). Choice of symmetry during refinement was dictated by 2D class averages (Fig 3, Appendix Fig S2). To test the symmetry, an initial 3D reference was built and refined without imposing any symmetry (C1), which resulted in clear six-pointed star reconstruction. Further refinements have been done utilizing C6 symmetry. The outer and lumenal diameters are 132 and 53 Å, and the height of the assembly is 38 Å. Molecular weight estimation from the size-exclusion chromatography (SEC) profile is ~600 kDa, which corresponds to twelve copies of TssA$_{VC}$ in one oligomer (634 kDa from sequence) (Appendix Fig S3).

The model built in cryo-EM density of TssA$_{VC}$ is composed of twelve C-terminal subunits (G376-T466, helices α8–α12) tightly packed as two interpenetrating rings, similar to *A. hydrophila* TssA (TssA$_{AH}$), but different from head-to-head stacked hexameric rings of the TssA$_{EC}$ assembly (Fig 3A and B, Appendix Fig S2) (Zoued *et al*, 2016; Dix *et al*, 2018). Each of the C-terminal subunits forms two interfaces with its neighbors. The first 688 Å$^2$ PISA-predicted interface is based on three conserved residues (W454, E455, and P456) from α11–α12 linker of one subunit and helix α10 from the neighboring subunit (Fig 3B). Phylogenetic analyses have shown

**Figure 3.  Cryo-EM of TssA$_{VC}$.**

A    Top view and side cutaway view of the TssA$_{VC}$ cryo-EM reconstruction, shown low-pass filtered at a lower (white, transparent) and higher (royal blue, non-transparent) threshold. Possible (−60°; 60°) range of motion of Nt2 domain relative to the CTD ring plane is shown.

B    Top view of the ribbon diagram of TssA$_{VC}$ CTD model. Two interfaces are highlighted with green circles. Enlarged side view of interface between helices α9–α11 of two neighboring subunits (top right), enlarged side view of interface between α11–α12 linker of one subunit and helix α10 from the neighboring subunit based on conserved WEP motif (bottom right). Part of Nt2-CTD linkers are highlighted with red circles.

C    Top view (XY-plane) and side view (XZ-plane) of the TssA$_{VC}$ Nt2 domain cryo-EM reconstruction (pink, non-transparent) shown with fitted Nt2-dimer model (left). Partial Nt2-CTD and Nt1-Nt2 linker densities are highlighted with black arrows. Top and side views of ribbon diagram of Nt2 dimer (middle). Representative 2D class averages of TssA$_{VC}$ particles with visible connections between Nt2 dimer and CTD (top right), representative 2D class averages of Nt2-dimer particles, Nt2-CTD linker and Nt1-Nt2 linker densities are highlighted with blue and yellow arrows (bottom right).

D    Side and top views of putative model of TssA$_{VC}$ Nt2-CTD ring fitted into cryo-EM reconstruction (royal blue, non-transparent) of the distal end of TssA$_{VC}$ T6SS (EMD-3878). TssA$_{VC}$ Nt2-CTD symmetrized reconstruction shown (white, transparent), fitted Nt2-CTD model shown as a ribbon diagram.

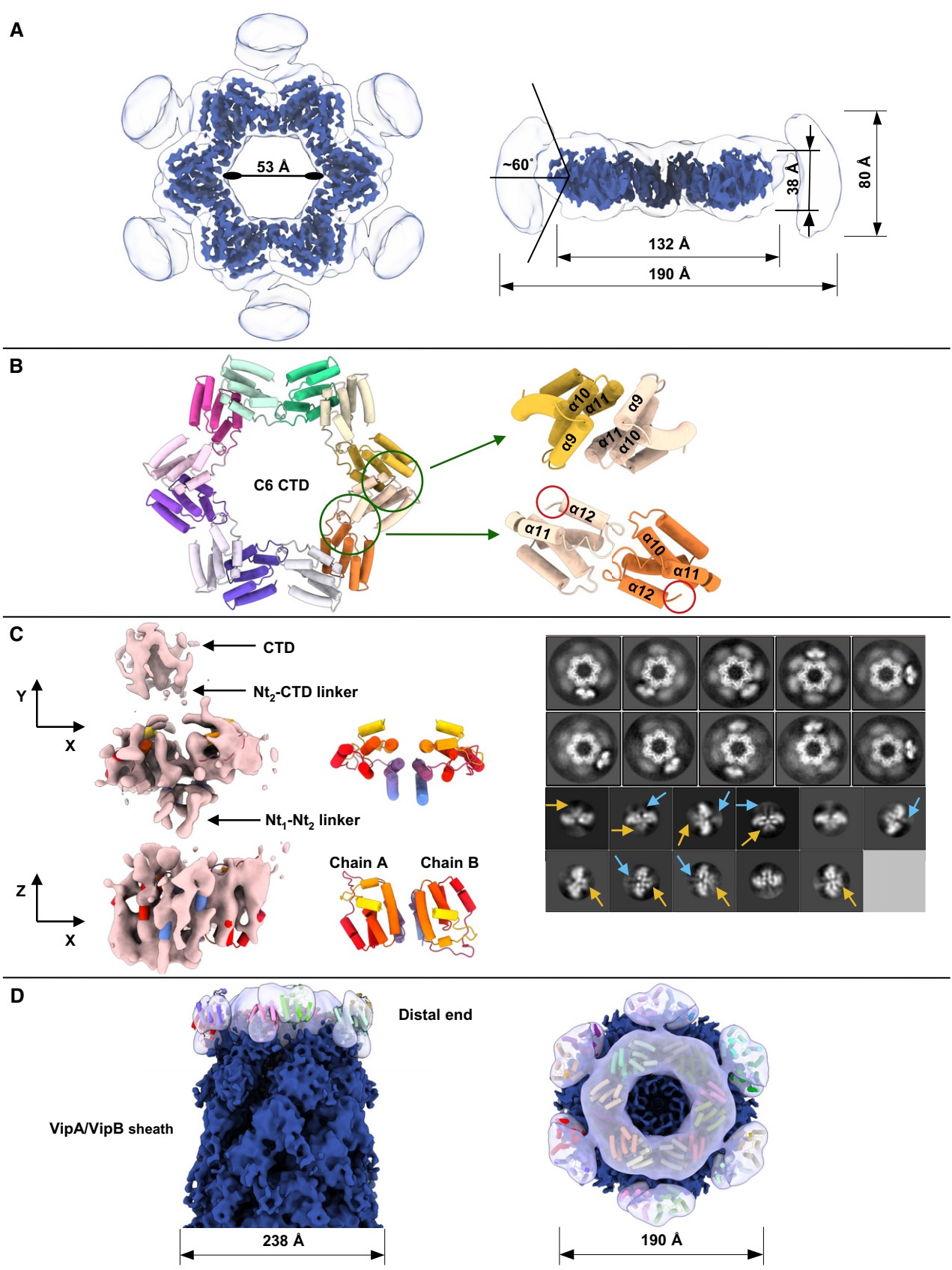

Figure 3.

that the WEP motif is only present in TssA proteins of Class A (Dix *et al*, 2018). Second 672 Å$^2$ interface forms a sixfold axis between helices α9–α11 of two neighboring subunits (Fig 3B). Well-resolved TssA$_{VC}$ star-like CTD is surrounded by disordered density (Fig 3A–D), which most likely represents the middle N-terminal domain (Nt2). 2D class averages show the presence of dimers (most likely Nt2 dimers) connected to the CTD ring (Fig 3C). Seven helices α1–α7 constituting each Nt2 subunit (D218-S363) are visible in 2D class averages, as well as two ~27 residues Nt2-CTD interdomain linkers (Fig 3C). Due to linker mobility and length, Nt2 dimers are not dominated by sixfold symmetry of the CTD assembly. Instead, they show moderate degree of motion relative to the sixfold axis and degree of rotation around linker axis (Fig 3A and C). Similar behavior of Nt2 dimers was observed for crystal packaging of Nt2-CTD TssA$_{AH}$ (Dix *et al*, 2018).

To improve the resolution of the density corresponding to the Nt2 dimer, symmetry of TssA$_{VC}$ reconstruction was relaxed from C6 to C1 and focused classification and refinement were performed. Resulting 3D reconstruction was used for picking template creation. Picked particles after standard processing pipeline were reconstructed to 6.6 Å resolution (Appendix Fig S2F and G). A homology model of TssA$_{VC}$ Nt2 domain was fitted and refined into cryo-EM reconstruction using molecular dynamics flexible fitting (MDFF) (Fig 3C). Cryo-EM density outlines all helices α1–α7 of both Nt2 subunits, assembled into a dimer with 591 Å$^2$ interface between helices α1–α3. In addition, partial densities of Nt2-CTD and Nt1-Nt2 linkers are present in the reconstruction (Fig 3C). Based on this, we predict that the Nt1 domain is on the periphery of the CTD-Nt2 assembly, however, likely connected through flexible linkers and thus not resolved in our cryo-EM structure. Our reconstruction of TssA$_{VC}$ CTD fits as a rigid body with CC = 0.87 into the distal end of VipA-N3 sheath mutant reconstruction (Fig 3D).

## TagA$_{VC}$ stabilizes extended sheath and prevents its excessive polymerization

Class B of TssA-like proteins is represented by *V. cholerae* TagA (VCA0121) and the recently characterized *E. coli* TagA (Santin *et al*, 2018). TagA$_{VC}$ shares 19.3% sequence identity with TagA$_{EC}$ (Appendix Table S1). To test the role of TagA$_{VC}$ on sheath dynamics, we first deleted *tagA$_{VC}$* and imaged sheath assembly. Manual inspection of about 250 cells imaged for 5 min showed that the deletion strain had fivefold decreased number of sheath structures per cell. Overall sheath contractions per cell per minute decreased by threefold when compared to the parental strain (Figs 1A and 4A, Movie EV6 and Fig EV3D). This suggests that sheaths in Δ*tagA$_{VC}$* strain are less stable overall, in particular that the sheaths contract soon after full assembly, and thus, fewer fully assembled sheaths are detected in these cells (Fig EV3A and B). Indeed, in a bacterial competition assay the Δ*tagA$_{VC}$* mutant showed killing kinetics similar to the parental strain (Fig EV1B left). It is likely that the very high overall T6SS activity in *V. cholerae* compensates threefold reduction in contractions per minute in the Δ*tagA$_{VC}$* strain, and thus, no difference in killing of prey cells is observed using this assay. Therefore, we compared the Δ*tssE* mutant, a strain that is significantly less efficient in killing *E. coli* cells, to the strain lacking both *tssE* and *tagA$_{VC}$* in bacterial competition

assay performed with higher ratio of *V. cholerae* to *E. coli* (Fig EV1B right). We observed slower killing kinetics in the double mutant showing that TagA$_{VC}$ contributes to T6SS function. In both the parental strain and the Δ*tssE* background strain, the presence of TagA slightly increases speed of sheath polymerization (Fig EV1C).

Importantly, we noticed that the lengths of T6SS sheaths are significantly increased in Δ*tagA$_{VC}$* mutant (Fig EV3E). In the parental strain, sheath assembly is initiated at the membrane and continues until the opposing side of the cell. On the other hand, sheaths in the Δ*tagA$_{VC}$* strain often continue polymerizing even after reaching the opposite side of the cell. This leads to extension along the longitudinal axis of the cell until reaching the cell pole (Fig 4A–C). This continued sheath polymerization results in various amount of sheath bending. Such bent sheath structures are not readily observed in the parental strain, suggesting that TagA$_{VC}$ could interact with the extended sheath and prevent further polymerization upon reaching membrane opposite to the site of assembly initiation.

Interestingly, while most sheath structures fully extend and contract normally despite significant bending, we also observed that during polymerization sheaths in Δ*tagA$_{VC}$* mutant cells may apparently detach from their membrane anchor (Fig 4C). Surprisingly, this did not result in immediate sheath contraction and the sheaths further extended to the whole cell length before subsequent contraction (Fig 4C), suggesting that those structures still harbored an intact baseplate. Contraction of such a detached sheath structure most likely failed to secrete proteins unless it reattached to another trans-envelope complex. Altogether, the absence of *tagA$_{VC}$* resulted in sheath bending as well as detaching from its membrane anchor, which are likely the main reasons for lower overall T6SS activity. The observed phenotypes are consistent with the observations made recently for TagA$_{EC}$ (Santin *et al*, 2018).

## TagA$_{VC}$ localizes to membrane and prevents sheath assembly upon overexpression

Analysis of TagA$_{VC}$ sequence suggested that the protein may localize to the membrane (Fig EV1A) similarly to TagA$_{EC}$ (Santin *et al*, 2018). Since the overall signal of chromosomally expressed TagA-mNeonGreen was low, we decided to first overexpress the TagA$_{VC}$ protein. We tested if TagA$_{VC}$ overexpression changes T6SS activity and found that overexpression of TagA$_{VC}$ led to complete inhibition of T6SS-dependent killing of *E. coli*, while a strain overexpressing TssA$_{VC}$ was still able to kill like the parental strain (Fig EV4A). Interestingly, fluorescence microscopy revealed that no sheaths were assembled when TagA$_{VC}$ was overexpressed (Fig EV4B). In addition, overexpressed TagA$_{VC}$-mNeonGreen localized to the membrane and inhibited sheath assembly, suggesting that TagA$_{VC}$-mNeonGreen has the same properties as untagged protein (Fig EV4B). In contrast, overexpression of TssA$_{VC}$-mNeonGreen had no influence on sheath assembly (Fig EV4C). When we removed the inducer after prolonged overexpression, we observed that during the next 2 h the cells that overproduced TssA$_{VC}$-mNeonGreen displayed regular T6SS activity, while the cells that overproduced TagA$_{VC}$-mNeonGreen only slowly regained T6SS activity (Fig EV4C). Interestingly, when TagA$_{VC}$ was overexpressed in the

VipA-mCherry2 TssA$_{VC}$-mNeonGreen strain, the TssA$_{VC}$-mNeon-Green spots remained closely associated with the membrane and colocalized with sheath spots (Fig EV4D). High concentration of TagA$_{VC}$ thus results in retention of TssA$_{VC}$ at sheath assembly initiation sites and inhibition of sheath assembly.

## TagA$_{VC}$ forms a hexamer at the distal end of an assembled sheath

To further understand the function of TagA$_{VC}$ protein, we imaged localization of chromosomally expressed TagA$_{VC}$-mNeonGreen

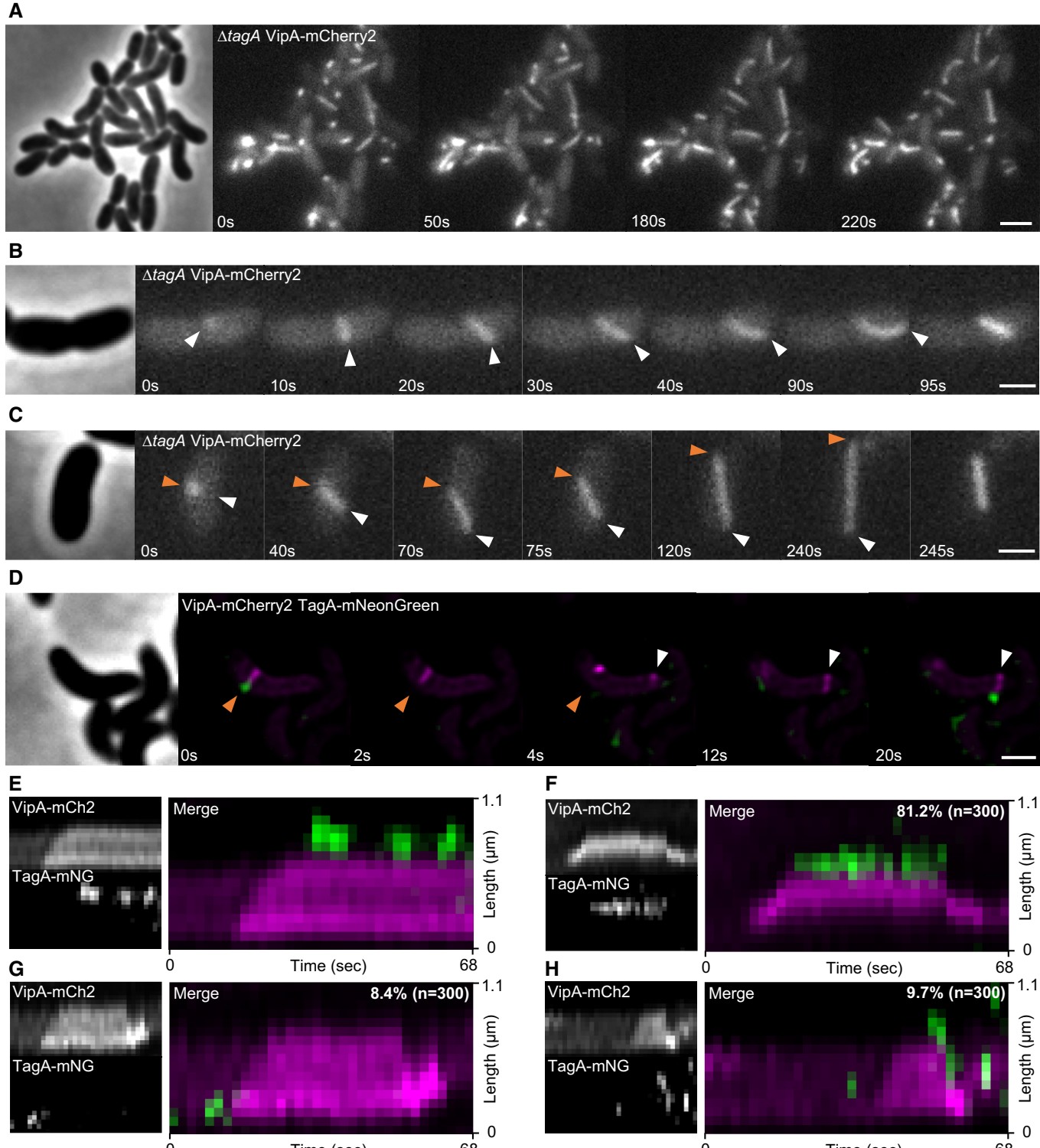

**Figure 4.**

**Figure 4. TagA$_{VC}$ localization and influence on sheath stability.**

A     The $\Delta tagA$ mutant displays long and bent but still dynamic structures. Scale bar: 2 μm.

B, C     Frames from time lapse movies showing sheath dynamics in the $\Delta tagA$ mutant. Sheath polymerization does not stop when the distal end reaches opposing membrane (white arrow). Continued polymerization causes the sheath to tilt and distal end to move toward the bacterial cell pole, ultimately effecting bending of the whole structure. (C) Sheath detaches from its membrane anchor (orange arrow), keeps extending to maximum cell length, and then contracts. Scale bars: 1 μm.

D     Time lapse series of T6SS activity in tagged strain (VipA-mCherry2 TagA$_{VC}$-mNeonGreen). Orange arrow indicates sheath distal end, and white arrow indicates initiation site of another T6SS sheath. Scale bar: 1 μm.

E–H     Kymographs of dynamics observed for TagA$_{VC}$. (E) TagA$_{VC}$ localizes at the distal end when sheath reaches the cell periphery. Signal of TagA$_{VC}$-mNeonGreen disappeared and reappeared on this site repeatedly. (F) TagA$_{VC}$ localizes at the distal end when sheath reaches the cell periphery and stays associated with this site until contraction. (G) TagA$_{VC}$ briefly localizes at T6SS initiation site. (H) TagA$_{VC}$ briefly localizes at T6SS initiation site and at distal end, where it stays attached after contraction.

during sheath assembly. Localized TagA$_{VC}$-mNeonGreen signal, stable for at least two subsequent frames (2s apart), was occasionally detected near a sheath structure. In over 80% of these cases, TagA$_{VC}$-mNeonGreen localized at the site where the sheath distal end reached the cell periphery after full extension (Fig 4D–H, Movie EV7). Further analysis of time lapse movies revealed several distinct dynamics of TagA$_{VC}$ localization. In most cases, TagA$_{VC}$-mNeonGreen signal disappeared very briefly before sheath contraction (< 2 s) (Fig 4D). We also observed that TagA$_{VC}$ displayed a rather dynamic localization at the distal end of the sheath by appearing and disappearing periodically (Fig 4E). We cannot exclude, however, that photobleaching occurred or TagA$_{VC}$ spot shifted out of focus. Disappearance of TagA$_{VC}$ signal was not immediately followed by sheath contraction in these cases. Further, we found that in about 10% of the cases TagA$_{VC}$ localized to sheath assembly initiation sites but remained there only for few seconds (2–6 s) prior to sheath assembly (Fig 4G). In about 10% of the cases, we detected TagA$_{VC}$ on the distal end of contracted structures (Fig 4H). The fact that many sheaths contract soon after assembly (Fig 2B) likely explains the overall low frequency of formation of stable TagA$_{VC}$-mNeonGreen spots. In general, the observed dynamics of TagA$_{VC}$ is consistent with dynamics and localization of TagA$_{EC}$ (Santin *et al*, 2018).

To estimate the oligomerization status of TagA$_{VC}$, we used the same approach as described above and compared brightness of TagA$_{VC}$-mNeonGreen spots to the brightness of LacI-mNeonGreen spots bound to *lacO* arrays of different length (Fig EV2C and D). TagA$_{VC}$-mNeonGreen spots were as bright as LacI-mNeonGreen spots on 3× *lacO* array, which was near the detection limit of our approach. This suggests that TagA$_{VC}$ forms a hexamer (Fig EV2D).

### Interaction partners of TssA and TagA in *V. cholerae*

We noticed that the fraction of sheaths with TssA$_{VC}$-mNeonGreen spots still attached to sheath distal end after contraction increased from 3% in the parental strain to 30% in the $\Delta tagA_{VC}$ mutant (Figs 2E and EV3C). Together with the sheath assembly phenotype of the $\Delta tagA_{VC}$ strain and the TagA$_{VC}$ localization data, this suggested a possible interaction between TagA$_{VC}$ and TssA$_{VC}$ as also shown for these proteins in *E. coli* (Santin *et al*, 2018). To identify interaction partners of TssA$_{VC}$ and TagA$_{VC}$, we first performed a pulldown experiment using hemagglutinin (HA)-tagged proteins as baits, and second, to confirm the identified interaction partners, we used bacterial two-hybrid system (Karimova *et al*, 2017). We identified sheath component VipB, TagA$_{VC}$, and ClpV as interacting proteins of TssA$_{VC}$ in both pulldown and bacterial two-hybrid system (Appendix Fig S4A and B). This is consistent with

observation for TssA$_{EC}$; however, unlike TssA$_{EC}$, which was shown to directly interact with Hcp and several baseplate components (Zoued *et al*, 2016), we found no such interaction for TssA$_{VC}$ using pulldown or bacterial two-hybrid assays (Appendix Fig S4A and B). Using TagA$_{VC}$-HA as bait revealed interaction with ClpV, sheath components VipA and VipB as well as baseplate component TssK (Appendix Fig S4A). Bacterial two-hybrid assay confirmed TssK interaction; however, interactions with sheath subunits were not confirmed (Appendix Fig S4B).

### Baseplate localization of TssA1 in *P. aeruginosa*

To investigate the role of Class C TssA-like proteins, we turned to TssA1 in H1-T6SS of *P. aeruginosa*. It was reported earlier that deletion of *tssA1* in *P. aeruginosa* PAK led to severe decrease in T6SS activity (Planamente *et al*, 2016). However, T6SS activity was not completely abolished, and thus, we decided to investigate the role of TssA1 in the closely related strain *P. aeruginosa* PAO1 using fluorescence microscopy. We used a $\Delta retS$ mutant strain background, since RetS negatively affects H1-T6SS expression (Mougous *et al*, 2006). In the $\Delta retS$ TssB1-mCherry2 strain, H1-T6SS shows very fast sheath dynamics as one cycle of T6SS extension and contraction usually takes only 5–20 s. Extending and contracting structures can be observed in up to 50% of cells (Fig 5A, Movie EV8, Fig EV5C). Deletion of $tssA1_{PA}$ decreased the amount of detected sheath structures, and the majority of observed H1-T6SS activity consisted of dynamic sheath spots, but extending and contracting sheath structures were also detectable (Fig 5A, Movie EV8, Fig EV5C). Importantly, deletion of $tssA1_{PA}$ had little to no influence on the speed of sheath assembly (Figs 5A and EV5D).

So far, only indirect evidence pointed to localization of TssA1$_{PA}$ at the baseplate (Planamente *et al*, 2016). Consequently, we attempted to complement the $\Delta tssA1_{PA}$ mutant strain with mNeonGreen-tagged TssA1$_{PA}$ to observe its localization and dynamics. Unfortunately, full complementation of T6SS dynamics in the strain harboring TssB1-mCherry2 fusion was only possible with untagged version of TssA1$_{PA}$ (Fig EV5A, B and E). Additionally, in the absence of the native copy of TssA1$_{PA}$, most detected dynamic structures (87%) contained only TssA1$_{PA}$-mNeonGreen (Fig EV5E), suggesting that the complex formed solely from tagged TssA1$_{PA}$-mNeonGreen fails to recruit TssB1-mCherry2.

We next expressed TssA1$_{PA}$-mNeonGreen in the presence of the native copy of TssA1$_{PA}$ in the strain harboring TssB1-mCherry2 fusion. Colocalization analysis showed that the majority (197 out of 225) of dynamic structures formed in such strain contained both TssA1$_{PA}$-mNeonGreen and TssB1-mCherry2 (Fig EV5E). In 163

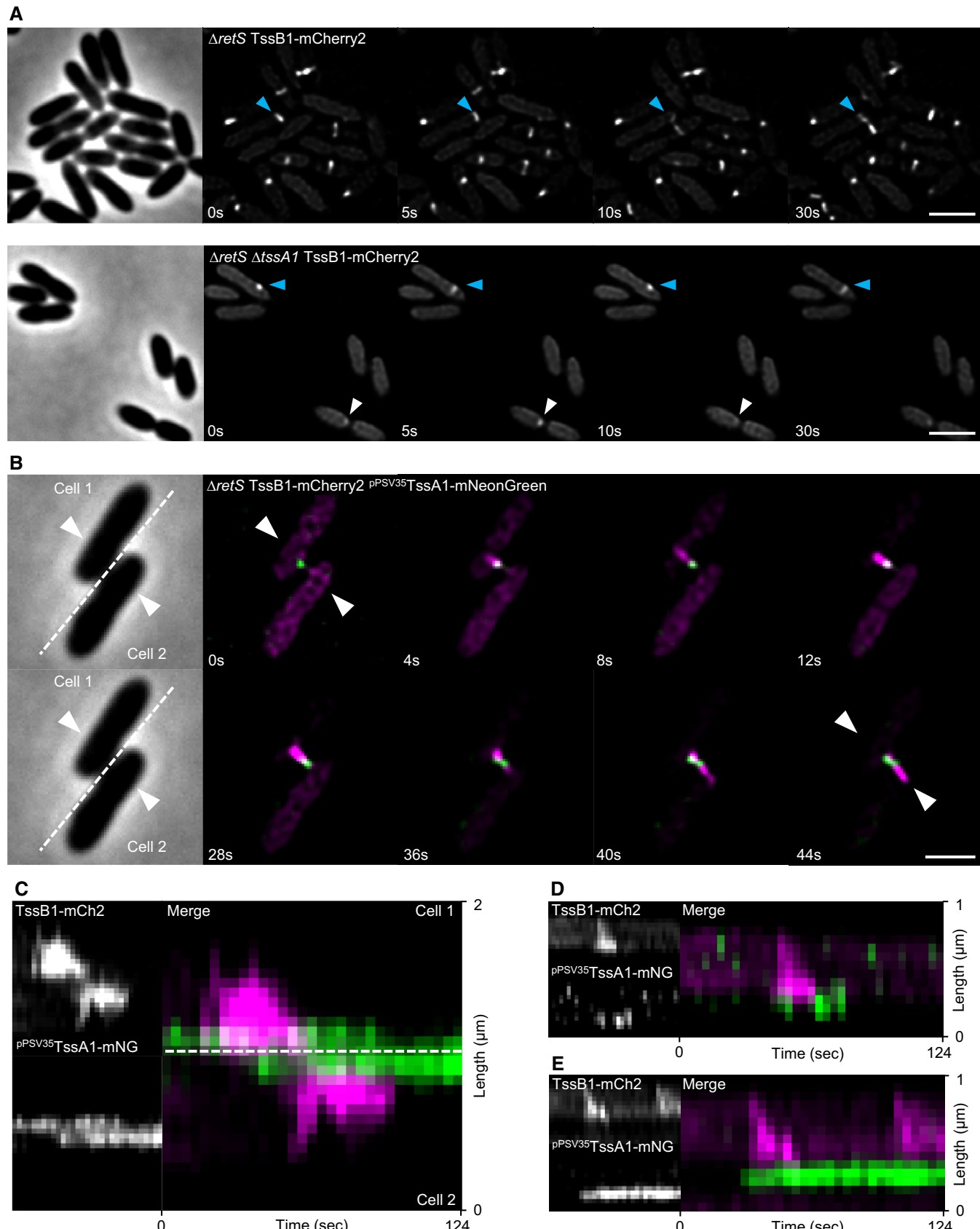

**Figure 5.**

**Figure 5.  TssA1$_{PA}$ localizes at T6SS initiation sites in *P. aeruginosa* PAO1.**

A   H1-T6SS dynamics in tagged parental strain ($\Delta retS$ TssB1-mCherry2, referred to as WT H1) and $\Delta tssA1_{PA}$ strain. Blue arrow indicates extending and contracting T6SS sheath structures, and white arrow points to dynamic sheath spots. Scale bars: 2 µm.

B   Fluorescence microscopy of T6SS dynamics in $\Delta retS$ TssB1-mCherry2 strain harboring pPSV35 plasmid with TssA1$_{PA}$-mNeonGreen fusion. Scale bar: 2 µm.

C–E   Kymographs of TssB1 and TssA1$_{PA}$ dynamics. TssA1$_{PA}$ localizes exclusively at sheath initiation site (B–E), disappears after contraction (C,D), or remains in place for longer timeframe (C,E). TssA1$_{PA}$ signal appears in neighboring cell shortly after T6SS attack (B,C).

cases, a TssA1$_{PA}$-mNeonGreen spot colocalized with a site from which an extended sheath assembled (as shown in Fig 5B). In 34 cases, the TssA1$_{PA}$-mNeonGreen spot colocalized with a dynamic sheath spot. Similar ratio between formation of dynamic sheath spots and contractile sheaths was observed for the cells expressing only the native copy of TssA1$_{PA}$ (Fig EV5C).

In the presence of native TssA1$_{PA}$, the tagged protein TssA1$_{PA}$-mNeonGreen formed a spot at the membrane before sheath signal appeared (Fig 5B–E, Movie EV9), which was followed by sheath extension away from this spot and later sheath contraction. Interestingly, after sheath contraction, TssA1$_{PA}$ signal disappeared (Fig 5D) or stayed in place for another round of sheath assembly and contraction from the same site (Fig 5E). During the dueling behavior of *P. aeruginosa* H1-T6SS, a new spot of TssA1$_{PA}$-mNeonGreen appeared in a cell within few seconds after a T6SS attack from the neighboring cell and before a new sheath assembled to carry out the counterattack (Fig 5B and C). Overall, these results suggest that TssA1$_{PA}$ is localized at the baseplate and increases the rate of sheath assembly initiation but likely plays no role in sheath polymerization. However, since the tagged TssA1$_{PA}$ is only functional in the presence of the untagged native copy of TssA1$_{PA}$, we cannot rule out a dual role of TssA1$_{PA}$ protein.

## Discussion

Here, we show that proteins with a conserved N-terminal domain PF06812 cluster into three main classes and mainly differ in their C-terminal sequence. These proteins are usually named either TssA or TagA or ImpA_N containing proteins (Mintz & Fives-Taylor, 2000; Zoued *et al*, 2017). One class of TssA proteins contains recently characterized TssA from *E. coli* (Zoued *et al*, 2016). This protein forms a dodecamer *in vitro*, interacts with TssK, TssE, VgrG, Hcp, TssB, and TssC, and localizes to the distal end of an assembling sheath (Zoued *et al*, 2016). Interestingly, Zoued *et al* showed that the N-terminal part of TssA$_{EC}$ containing ImpA_N domain might be responsible for interaction with sheath and to a lesser extent with baseplate components, while the C-terminus preferably interacted with Hcp. The TssA protein of *V. cholerae* is from the same class, and we show here that it plays the same role in T6SS biogenesis. Similarly to TssA$_{EC}$, most of the time TssA$_{VC}$ localizes to the distal end of assembling sheaths (Fig 2B). In addition to our high-resolution dodecameric cryo-EM structure, we show that intensity of one TssA$_{VC}$-mNeonGreen spot likely represented 12 molecules of mNeonGreen, suggesting that TssA$_{VC}$ forms a dodecamer during sheath assembly (Fig EV2D). Our photobleaching experiment suggests that once this dodecamer forms, it is stable during the whole sheath assembly process (Appendix Fig S1). However, TssA$_{VC}$ is not strictly required for sheath assembly as *tssA$_{VC}$*-negative strain still kills prey cells, assembles short dynamic sheaths and

occasionally also full-length sheath with significantly reduced assembly speed (Fig EV1). The H2-T6SS cluster in *P. aeruginosa* also encodes a TssA protein of the same class. Live-cell imaging of the sheath assembly and TssA2$_{PA}$ localization confirmed that also this protein localizes to the distal end of the assembling sheath (Fig 2). Taking these observations together, it is very likely that all TssA proteins of the Class A (Fig EV1) perform the same function in T6SS biogenesis and are required for fast and efficient polymerization of T6SS sheath. Interestingly, elongation of sheath of bacteriophage T4 does not require the action of chaperones (Arisaka *et al*, 1979). Therefore, while potentially not being strictly essential, Class A TssAs act as chaperones assisting sheath-tube copolymerization, which likely requires assistance in proper formation of the sheath hand-shake domain (Ge *et al*, 2015; Kudryashev *et al*, 2015; Wang *et al*, 2017). We thus propose to rename this class of TssA proteins to TsaC (T6SS sheath assembly chaperone).

TagA protein encoded by *V. cholerae* is a representative of the ImpA_N domain containing class B proteins (Fig EV1) that were initially proposed to have only accessory role in T6SS biogenesis (Zoued *et al*, 2017). A recent study provided evidence that a TagA protein from *E. coli* localizes to the membrane at the distal end of the sheath and holds it in place (Santin *et al*, 2018). Similarly, we show that TagA$_{VC}$ localizes to the cell periphery and most of the times forms hexameric spots at the distal ends of fully assembled sheaths (Fig 4). Deletion of *tagA$_{VC}$* resulted in about fivefold decrease in number of sheath assemblies, but only about threefold decrease in sheath contractions, and thus, the influence on the overall ability of *V. cholerae* to kill prey cells was negligible (Fig EV1B). Consistently with observations made in *E. coli* (Santin *et al*, 2018; Szwedziak & Pilhofer, 2019), the sheaths in *tagA$_{VC}$*-negative strain were mostly very long, stretching from one pole of a cell to another. In addition, the assembling sheaths were often bent and occasionally consequently detached from their membrane anchor (Fig 4). This suggests that TagA$_{VC}$ prevents extensive sheath polymerization, which may result in bending and sheath instability. Furthermore, sheaths in *tagA$_{VC}$* knockout mutant did not stay extended or anchored to the membrane as observed in the parental strain (Fig EV3A and B), which was also observed in *E. coli* (Santin *et al*, 2018). Interestingly, Szwedziak and Pilhofer showed recently that TagA$_{EC}$ is also required for fast sheath assembly as the *tagA$_{EC}$*-negative cells assembled the sheaths about 7-fold slower than wild type. In addition, electron cryotomography showed that the sheaths in *E. coli* are attached at both ends to the cell membrane, one end is connected to a baseplate and the second end to an uncharacterized structure likely composed of TagA$_{EC}$ and potentially other proteins (Szwedziak & Pilhofer, 2019). Moreover, the TagA$_{EC}$-mediated membrane attachment was also required for non-canonical sheath contractions observed in one-third of the contraction events where the full-length sheath or its part contracted away from the baseplate toward the distal end (Szwedziak & Pilhofer, 2019). However,

whether such non-canonical contractions lead to protein secretion and delivery to target cells remains to be shown.

We show that $TagA_{VC}$ interacts with $TssA_{VC}$, TssK, ClpV, VipA, and VipB. Interestingly, we observed that $TssA_{VC}$ spots disappeared from the fully extended sheaths (Fig 2C). Additionally, in the $tagA_{VC}$ mutant strain harboring VipA-mCherry2 and $TssA_{VC}$-mNeonGreen fusions, $TssA_{VC}$ often remained attached to the distal end of sheath even after contraction (Fig EV3C). This indicates that $TagA_{VC}$ blocks function of $TssA_{VC}$ potentially by removing $TssA_{VC}$ from the fully assembled sheath. Removal of $TssA_{VC}$ stops or slows down further sheath polymerization and thus stabilizes the extended sheath (Fig EV3A and B and summarized in Fig 6A). Moreover, when we overexpressed $TagA_{VC}$, sheath assembly was completely blocked and only spots of sheath signal colocalizing with $TagA_{VC}$ spots were detectable in the cells (Fig EV4D). This suggests that $TagA_{VC}$ may inhibit $TssA_{VC}$ function both during initiation of sheath assembly and after its extension. This could be mediated by TssA C-terminus since deletion of $TagA_{EC}$ and deletion of C-terminal part of $TssA_{EC}$ had similar effect on sheath dynamics (Szwedziak & Pilhofer, 2019). We propose to rename the class of TagA proteins (Fig EV1) to TsmA (T6SS sheath membrane anchor).

Interestingly, many T6SS clusters lack genes encoding TagA-like proteins, notably the H1-T6SS of *P. aeruginosa*. The sheath dynamics of this cluster is, however, very different from dynamics of sheath assembly of TagA-containing *E. coli* or *V. cholerae* clusters. Even though a model excluding sheath contraction was proposed for H1-T6SS (Corbitt *et al*, 2018), we show here that H1-T6SS sheaths assemble very quickly within 5-20s and immediately contract (Fig 5), which makes it technically challenging to detect the whole sheath assembly, contraction, and disassembly cycle. We also often observed dynamic sheath spots for H1-T6SS, which could be functional contractile sheath assemblies shorter than the diffraction limit of the used fluorescence microscopy technique. It was

shown previously that short sheath structures formed upon limitation of Hcp protein availability can still deliver effectors into target cells (Vettiger & Basler, 2016). This is also consistent with the observation that the $tssA_{VC}$-negative strain mostly forms small dynamic sheath spots but is still capable of prey killing, albeit with much lower efficiency than the parental strain. It is therefore possible that certain sheath-tube assemblies evolved to be more prone to contraction and contract either before or right after reaching the opposite side of the cell similarly to what was observed in *Acinetobacter baylyi* (Ringel *et al*, 2017). This could be achieved by evolution of less stable baseplate or membrane complex that may trigger sheath contraction before extending excessively and potentially bending and detaching from the cell envelope. On the other hand, the H2-T6SS cluster apparently lacks a homolog of TagA protein while the sheaths are very stable in the extended form. Those sheaths do not bend or extend from bacterial pole to pole as in the case of *tagA*-negative *V. cholerae* T6SS. Therefore, there might be yet another TagA-independent mechanism that stabilizes properly extended sheaths or this could be a consequence of a very slow polymerization speed of H2-T6SS sheaths (Fig 2F).

$TssA1$ protein of *P. aeruginosa* H1-T6SS cluster (Fig EV1, Class C) was shown to bind TssK1, TssF1, ClpV1, Hcp1, and TssB1, form ring-like structures, and localized to ends of contracted sheaths and thus was suggested to be a baseplate component (Planamente *et al*, 2016). Here, we show that $TssA1_{PA}$-mNeonGreen indeed localized to sheath assembly initiation sites (summarized in Fig 6B). When we expressed $TssA1_{PA}$-mNeonGreen in the absence of the native copy of $tssA1_{PA}$, the majority of $TssA1_{PA}$-mNeonGreen spots localized to the membrane and disappeared without subsequent sheath formation (Fig EV5E). However, in the strain harboring the native copy of $tssA1_{PA}$, localization of $TssA1_{PA}$-mNeonGreen at the membrane was generally followed by sheath assembly (Fig EV5E).

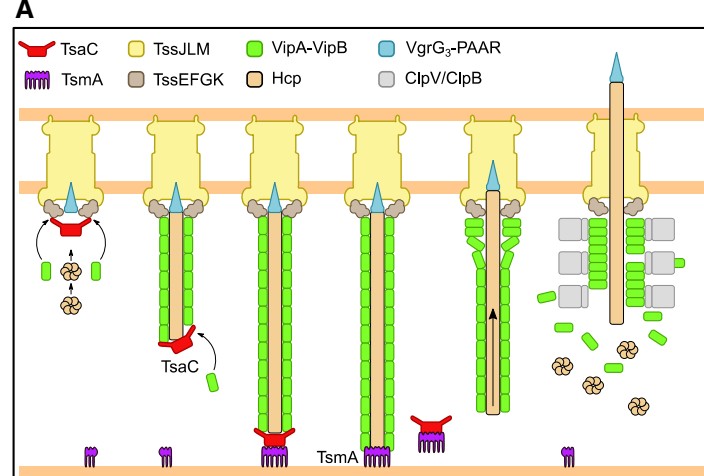
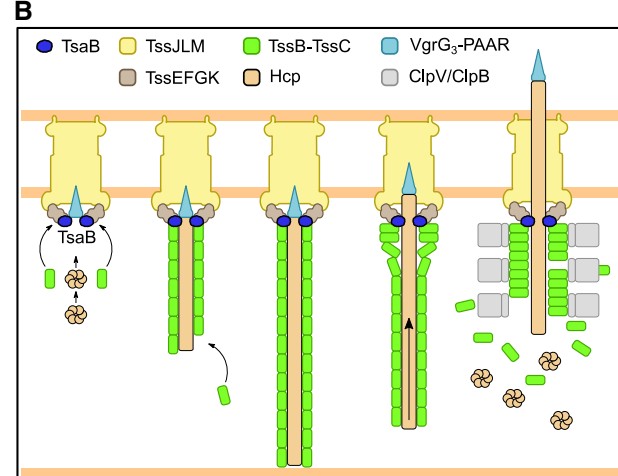

**Figure 6. Role of TsaC, TsmA, and TsaB in T6SS dynamics.**

A  Model of T6SS dynamics. After membrane complex has formed, TsaC assists in recruitment of baseplate components and sheath-tube copolymerization. Once extending sheath-tube has reached opposing cell membrane, TsmA replaces TsaC at the distal end and holds it in place. Shortly after TsmA disappearance, sheath contracts and releases spike tube to the environment. Sheath components are then recycled by the ClpV ATPase.

B  TsaB localizes at the baseplate and assists in the initiation of T6SS biogenesis.

Because both native and tagged TssA1$_{PA}$ were present in the cells, the oligomeric complex forming the TssA1$_{PA}$-mNeonGreen spots was likely composed of both proteins and such complex was thus able to initiate sheath assembly. It is important to note that nearly all dynamic sheath structures detected in the strain expressing both native and tagged TssA1$_{PA}$ proteins colocalized with TssA1$_{PA}$-mNeonGreen spots. Therefore, it is unlikely that TssA1$_{PA}$-mNeon-Green spot formation and sheath assembly initiation are two separate events or that the TssA1$_{PA}$ spot formation is an artifact of tagging by mNeonGreen. When we deleted *tssA1$_{PA}$* and analyzed the sheath assembly dynamics, we observed dramatic decrease in the number of detectable sheath assemblies (Fig 5). However, unlike deletion of *tssA$_{VC}$*, removing *tssA1$_{PA}$* had only subtle influence on sheath assembly speed (Figs 5 and EV5D). This suggests that TssA1$_{PA}$ is required only for sheath assembly initiation and raises the question whether there are other proteins encoded by H1-T6SS that may chaperone sheath-tube assembly. We propose renaming the class of proteins similar to TssA1$_{PA}$ to TsaB (T6SS sheath assembly baseplate).

In summary, we show here that sheath-tube assembly is a highly coordinated process, which requires various proteins sharing a conserved N-terminal domain. The precise understanding of the roles of these proteins will require high-resolution structures of their complex with the sheath or baseplate as well as structure-based mutagenesis followed by detailed analysis of sheath dynamics.

# Materials and Methods

### Strains and culture conditions

*Vibrio cholerae* 2,740–80 and *P. aeruginosa* PAO1 Δ*retS* fluorescent protein-tagged strains as well as knockout strains were generated as described previously (Basler & Mekalanos, 2012). Recombinant clones were checked by colony PCR and sequence-verified. *Escherichia coli* MG1655 (*lacZ*+) was used as prey strain in bacterial competition assays. Bacterial two-hybrid strains were generated using the instructions from the manufacturer (Euromedex). For bacterial two-hybrid assay, cells were incubated at 30°C for 24–48 h according to instructions. Stains used in this study are listed in Appendix Table S2. Bacteria were grown in LB broth at 37°C aerobically. Ampicillin (300–500 μg/ml), streptomycin (100 μg/ml), kanamycin (50 μg/ml), irgasan (20 μg/ml), and gentamicin (15 μg/ml) were used as supplements in growth media when necessary.

### Bacterial competition assay

*Vibrio cholerae* can lyse or permeabilize *E. coli* (MG1655, *lacZ*+) with its active T6SS, indicated by conversion of a membrane impermeable β-galactosidase substrate (chlorophenol red-β-D-galactopyranoside, CPRG) over time. Strains were mixed in a ratio of 1:5 or 2:1 (*V. cholerae*: *E. coli*) and co-incubated on solid LB agar supplemented with 40 μg/ml CPRG and 300 μg/ml Ampicillin when strains harboring pBAD24 plasmid were used. Killing of *E. coli* prey cells was recorded by monitoring OD$_{572}$ increase on a 96-well plate reader. At least three independent experiments were run on different days. Average of these experiments is shown.

### Image analysis

ImageJ was used to measure frequency of contractions and length of WT and Δ*tagA* mutant sheath structures before contraction. Sample images were taken from at least two different days of acquisition and three different time points within a time lapse series.

### Fluorescence microscopy

Fluorescence microscopy and processing of images were done as described in Brackmann *et al* (2017). Briefly, overnight cultures were washed with LB medium, diluted 1:100, supplemented with antibiotics, and cultivated to an optical density (OD) at 600 nm of 1. Then, cells were concentrated to OD 10, spotted on a LB 1% agarose pad, and covered with a glass coverslip. Bacteria were either directly imaged at 30°C or directly incubated on the agarose pad for 45 min prior to imaging (*P. aeruginosa* PAO1). We used a Nikon Ti-E inverted motorized microscope with Perfect Focus System and Plan Apo 100× Oil Ph3 DM (NA 1.4) objective lens. The microscope was equipped with SPECTRA X light engine (Lumencor), and ET-EGFP (Chroma #49002) and ET-mCherry (Chroma #49008) filter sets were used to excite and filter fluorescence. The setup further contained a sCMOS camera pco.edge 4.2 (PCO, Germany) (pixel size 65 nm) and VisiView software (Visitron Systems, Germany) to record images. Temperature control was set to 30°C, and humidity was regulated to 95% by an Okolab T-unit (Okolab). Additional image processing was carried out using Fiji (Schindelin *et al*, 2012) as described previously (Basler *et al*, 2013). For photobleaching, mNeonGreen fluorescence was reduced using a VS-AOTF 488 nm Laser system mounted with iLas2 laser merge on the microscope, allowing simultaneous laser and LED illumination. Photobleaching experiments were carried out as described earlier (Vettiger *et al*, 2017). Image series of all experiments with TssA- or TagA-mNeonGreen fusions as well as all *P. aeruginosa* image series were subjected to deconvolution. Deconvolution was carried out using Huygens Remote Manager (http://huygens-rm.org). Background estimation was set to auto, 40 iterations were run, quality change stopping criterion was 0.1, and deconvolution algorithm used was classic maximum-likelihood estimation.

### Fluorescence quantification

*lacO* arrays were designed to contain repeats of the operator sequence (aattgtgagcggataacaatt) with 15 random nucleotide spacers. LacI$_{351}$ lacking C-terminal tetramerization domain, which binds DNA like WT (Gregory *et al*, 2010) was fused to mNeonGreen and used to estimate number of mNeonGreen molecules needed to detect fluorescent spots of certain intensity in the cells. Raw mNeonGreen signal from LacI, TagA$_{VC}$, and TssA$_{VC}$ spots was collected using ImageJ. mNeonGreen signal of three consecutive images of LacI or TssA$_{VC}$/TagA$_{VC}$ spots (approximately 5 × 5 pixels) in a cell was averaged. Subsequently, an average signal of the cell cytosol of the corresponding frames was subtracted from the average signal of the spot and compared to random spots of similar size inside the cell. All measurements were done on unprocessed images.

## Co-IP and mass spectrometry

Co-IP was performed using Pierce Anti-HA Magnetic Beads. Cells were grown in LB medium to OD 600 nm of 1 with appropriate supplements, and HA-tagged protein expression was induced with 0.002% L-arabinose. Cells were lysed using lysozyme and CelLytic B (Sigma), and lysates were used directly for binding to Anti-HA Magnetic Beads. Binding and washing were done according to the recommended protocol (Pierce). Proteins were eluted by boiling at 98°C for 10 min and resuspended in TN-buffer, reduced, and alkylated. Samples were subsequently used for mass spectrometry analysis as described previously (Brackmann *et al*, 2017). Briefly, proteins were digested and supplemented with TFA to a final concentration of 1% overnight. Peptides were cleaned with PreOmics Cartridges (PreOmics, Martinsried, Germany) following the manufacturer's instructions. After drying under vacuum, peptides were resuspended in 0.1% aqueous formic acid solution at a concentration of 0.5 mg/ml. 0.5 µg of peptides of each sample was subjected to LC-MS analysis as described earlier (Brackmann *et al*, 2017). MS1 and MS2 scans were recorded at a target of 1E6 ions and 10,000 ions, respectively. One microscan was acquired for each spectrum, and collision energy was 35%. All raw data acquired by DDA were converted to mgf format (version 3.0, ProteoWizard, http://proteowizard.sourceforge.net/). Resulting files were searched against a decoy (consisting of forward and reverse protein sequences) database of predicted protein sequence of *V. cholerae* (UniProt, Organism ID: 243277, download date 11/09/2017, containing known contaminants) using Mascot (Matrix Science, version 2.4). Search parameters were as follows: Full tryptic specificity was required (cleavage after lysine and arginine residues unless followed by proline); up to three missed cleavages allowed; carbamidomethyl (C) was set as a fixed modification; oxidation (M) and acetyl (protein N-term) were set as variable modifications; 0.6-Da fragment mass tolerance for CID tandem mass spectra; and 10 ppm precursor mass tolerance. After importing results to Scaffold (http://www.proteomesoftware.com, version 4), FDR rate was set to < 1% for protein identifications by the local Scaffold FDR algorithm based on the number of decoy hits. Co-IP and mass spectrometry results are summarized in Appendix Table S3.

## *tssA*$_{VC}$ gene subcloning and protein expression

*tssA*$_{VC}$ (amino acids 1–469, Gene number: VC_A0119) of *V. cholerae* VC2740-80 was cloned into a modified pACEBACI (Geneva Biotech) expression vector containing a GATEWAY (Life Technologies) cassette with N-terminal His$_{10}$ tag according to the manufacturer's protocol. Bacmid and virus were generated in Sf21 cells (Expression systems) in Insect-Xpress medium (Lonza), following the MultiBac instructions. Mycoplasma contamination was tested using MycoAlert™ Mycoplasma Detection Kit (Lonza), and no contamination was detected. Protein expression was done according to MultiBac instructions. Cells were harvested 3 days after infection by centrifugation and stored at −80°C until further use.

## TssA$_{VC}$ protein purification

TssA$_{VC}$ protein was purified at 4°C. 40 g of cell pellet was resuspended in 160 mL of lysis buffer (50 mM Tris–HCl pH 7.5, 200 mM NaCl, 5 mM MgCl$_2$, 20 mM imidazole, 10% glycerol, 1× complete protease inhibitors, and 5 mM 2-mercaptoethanol) and passed 3 times on a microfluidizer operating at 15k psi. The cell lysate was clarified by centrifugation at 95k g for 1 h. The supernatant was filtered using 0.22-µm filters and applied to a 5 ml HisTrap HP affinity column (GE Healthcare) pre-equilibrated in lysis buffer. The column was washed with 10 CV of NiW buffer (50 mM Tris–HCl pH 7.5, 500 mM NaCl, 50 mM imidazole, 10% glycerol, and 5 mM 2-mercaptoethanol) and eluted with elution buffer (25 mM Tris–HCl pH 7, 100 mM NaCl, 750 mM imidazole, 5% glycerol, and 5 mM 2-mercaptoethanol). Wash and elution fractions were checked on an SDS gel (Appendix Fig S3A). Eluate fractions were pooled, mixed with TEV protease at 1:10 (m/m), and dialyzed overnight against gel filtration buffer (20 mM Tris–HCl pH 7.5, 150 mM NaCl, 5 mM MgCl$_2$, and 1 mM DTT) in 10 kDa MWCO membrane. TEV protease digested protein was concentrated using a Millipore 100 kDa centrifugal device and injected to a 10/300 Superose 6 increase column (GE Healthcare) equilibrated with gel filtration buffer. Peak fractions at 2 mg/ml were immediately used for cryo-EM grid preparation. Due to the preferential orientation of the particles in the vitreous ice, sodium cholate at CMC concentration was added. Detergent selection was monitored by tilt angular distribution of reconstructed volumes.

## Cryo-EM data acquisition

An aliquot of 3 µl of TssA$_{VC}$ sample was applied onto non-glow-discharged Lacey holey carbon grids (Carbon Cu 300 mesh, Ted Pella), blotted for 3 s, and vitrified using a Leica EM GP2 (Leica Microsystems). The chamber was kept at 20°C and 90% humidity during the blotting process. Images were collected on an FEI Titan Krios TEM (ThermoFisher Scientific), operated at 300 kV, and equipped with a Gatan Quantum-LS energy filter (zero loss slit width 20 eV; Gatan Inc.); data were recorded with a post-GIF K2-Summit direct electron detector (Gatan Inc.) in electron counting mode (60 e/Å$^2$ total dose, fractionated into 40 frames), at a calibrated pixel size of 1.058 Å. A total of 2,963 movies were collected using SerialEM (Mastronarde, 2005).

## Image processing and model building

### TssA$_{VC}$

Recorded movies were preprocessed online with FOCUS (Biyani *et al*, 2017). Movies were drift-corrected and dose-weighted using MotionCor2 (Zheng *et al*, 2017). The contrast transfer function of each micrograph was estimated using the CTFFIND4.1 (Rohou & Grigorieff, 2015). An initial particle set was picked using Gautomatch without templates (developed by Zhang, K., MRC Laboratory of Molecular Biology, Cambridge, UK) and then 2D classified in cryoSPARC v1 (Punjani *et al*, 2017). Particles selected from the best 2D classes were used for *ab-initio* reconstruction and subsequent 3D refinement to 10 Å resolution without imposing symmetry. Reference-based projections were generated, low-pass filtered to 20 Å using EMAN2 (Tang *et al*, 2007), and used as templates to pick 261,412 particles with Gautomatch. After several rounds of 2D classification in RELION3 (Zivanov *et al*, 2018), best 93,235 particles were selected from 2D class averages showing tilted and side views in addition to preferential top or bottom views. Consensus 3D

Johannes Paul Schneider et al

*The EMBO Journal*

refinement iterated with two rounds of 3D classification revealed the best class with well-resolved CTD density. A final 3D refinement with a soft-mask that included only CTD density, followed by post-processing resulted in a map with a global resolution of 3.9 Å (Appendix Fig S2). Global auto-sharpening using PHENIX (Afonine *et al*, 2018) was applied to the map after final 3D refinement, and resulted sharpened map was used for model building. Local resolution calculation was done using the ResMap web service (Kucukelbir *et al*, 2014), and 3D Fourier shell correlation was calculated with 3DFSC web service (Tan *et al*, 2017). One dimensional histogram for a tilt angle was plotted using plot_indivEuler_histogram_fromStarFile.py script (https://github.com/leschzinerlab/).

Homology model of CTD TssA$_{VC}$ was generated by SWISS-MODEL (Waterhouse *et al*, 2018) based on the X-ray structure of C-terminal domain from Nt2-CTD domains of TssA$_{AH}$ from Dix *et al* (PDB-6G7C). Initial rigid-body fitting was done with COLORES (Wriggers, 2012) and further refined with PHENIX real space refinement (Adams *et al*, 2010). Rounds of manual adjustment in Coot (Emsley *et al*, 2010) were iterated with structure reliability evaluation using MolProbity web service (Appendix Table S4) (Chen *et al*, 2010).

### TssA$_{VC}$ Nt2

Symmetry of the C6 reconstruction of TssA$_{VC}$ was relaxed to C1, and one of the six Nt2 dimers was focused 3D refined to 10 Å resolution. The resulting reconstruction was used for Gautomatch template creation. 1,184,380 particles were picked and 2D classified several times to clean the data set from false positives. Best 33,501 particles from 2D class averages with visible alpha helices were selected and processed first with RELION3, and further refined using non-uniform refinement in cryoSPARC v2, resulted in 6.6 Å resolution reconstruction. A homology model of Nt2 TssA$_{VC}$ was generated with SWISS-MODEL based on the X-ray structure of Nt2 domain from Nt2-CTD domains of Ah TssA from Dix *et al* (PDB-6G7C) and further refined with molecular dynamics flexible fitting (MDFF) and *phenix.real_space_refine* using Namdinator web service (Kidmose *et al*, 2019). To find relative orientations of Nt2 dimer and CTD ring, partial signal subtraction and 3D classification without alignment were used. Two masks were created, first around one of Nt2-dimers corresponding density and second around five symmetry-related copies. Symmetry of the C6 reconstruction of TssA$_{VC}$ was relaxed to C1, and five symmetry-related copies of Nt2 dimer were subtracted from all C6-C1 expanded particles, and 3D classified without alignment into ten classes using T-factor of 10. 3D class with improved Nt2-dimer density was selected, and corresponding particles were re-extracted without subtraction, and further 3D refined with local angular searches without imposing symmetry (C1) to 10 Å resolution. Resulted reconstruction was C6 symmetrized and rigid-body fitted into cryo-EM reconstruction of the TssA$_{VC}$ T6SS distal end (EMD-3878) (Fig 3D). Finally, previously reconstructed Nt2-dimer domain (Fig 3C) was rigid-body fitted into symmetrized reconstruction (Fig 3D).

## Data availability

- EM maps: Electron Microscopy Data Bank EMD-4898 (https://www.ebi.ac.uk/pdbe/emdb/).
- Electron micrographs: Electron Microscopy Public Image Archive EMPIAR-10271 (https://www.ebi.ac.uk/pdbe/emdb/empiar/).
- Atomic coordinates: Research Collaboratory for Structural Bioinformatics Protein Data Bank PDB 6RIU (http://www.rcsb.org).

**Expanded View** for this article is available online.

### Acknowledgements

The work was supported by Swiss National Science Foundation Grants BSSGI0_155778 and 31003A_159525 as well as the University of Basel. Calculations were performed at sciCORE (http://scicore.unibas.ch/) scientific computing core facility at University of Basel. We acknowledge the Biozentrum BioEM Lab for access to their microscopes, and Mohamed Chami and Lubomir Kovacik for support with cryo-EM data collection. We also thank Dr. Anna Hagmann and Prof. Timm Maier for their help with cloning and initial trials of TssA$_{VC}$ protein expression.

### Author contributions

JPS generated strains, performed experiments, and processed and analyzed data. SN and RA performed sample purification, SN performed cryo-EM data collection, processing, and model building. ML generated strains and performed bacterial two-hybrid experiments. PDR assisted in image data analysis. HS assisted and supported cryo-EM data collection. MB conceived the project and analyzed the data. JPS and MB wrote the manuscript. All authors read and approved the manuscript.

### Conflict of interest

The authors declare that they have no conflict of interest.

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
