## [Review Process File · The EMBO Journal]

Diverse roles of TssA-like proteins in the assembly of bacterial Type VI secretion systems

Johannes Paul Schneider, Sergey Nazarov, Ricardo Adaixo, Martina Liuzzo, Peter David Ringel, Henning Stahlberg and Marek Basler

Review timeline:

Submission date:	30th Sep 2018
Editorial Decision:	11th Jan 2019
Revision received:	29th Apr 2019
Editorial Decision:	24th May 2019
Revision received:	8th Jul 2019
Accepted:	16th Jul 2019

Editor: Ieva Gailite

Transaction Report:

1st Editorial Decision

11th Jan 2019

Thank you for submitting your manuscript for consideration by the EMBO Journal. I sincerely apologise for the unusual delay in the assessment of your work due to belated submission of referee reports. We have now received three referee reports on your manuscript, which are included below for your information.

As you will see from the comments, the reviewers appreciate the work and the topic. However, they also raise a number of issues (mainly clarification or further discussion) that have to be addressed, including inclusion of the meanwhile published manuscripts by Santin et al., 2018 Nat Microbiol and Dix et al., 2018 Nat Commun in the discussion. Based on the overall interest expressed in the reports, I would like to invite you to submit a revised version of your manuscript in which you incorporate the requested changes.

REFeree REPORTS:

Referee #1:

This manuscript describes the functions of TssA and TagA proteins in the type VI secretion system (T6SS). The authors perform thorough *in vivo* fluorescence experiments that clearly demonstrate functions of TssA and TagA. They also provide data showing which T6SS proteins are bound by TssA and TagA.

This is an interesting paper with well produced data. However, the paper is confusing in parts. In the bioinformatic figure, there are full strain names included and it is hard to connect those names with the proteins that are being discussed in the paper. The authors say that certain proteins are closely related, but it is impossible to know what this means. What are the pairwise percent identities between *Vibrio* and *E. coli*? Are all domains conserved? What is the percent identity with the *Pseudomonas* proteins. In the section discussing these proteins, Uniprot accession numbers are

confused with Pfam family names (e.g. line 128). I also found it difficult to understand what was being shown in the kymograph parts of Fig. 4 and Fig. 5. The authors need to explain this as not all readers are familiar with this type of figure.

The authors use a "very sensitive" assay for activity of the Vibrio T6SS. The authors should discuss whether this is too sensitive an assay. I wonder how mutations of genes like *tssE*, which are absolutely conserved, could not result in full loss of activity (e.g. knocking out the homologous phage gene completely eliminates tail production). Is *E. coli* too easy to kill? Should activity also be tested against a more challenging enemy (e.g. amoeba).

The conclusions concerning TssA function are not described very clearly. The authors mention results from other studies that suggest a variety of functions for TssA. If the authors have an opinion about what is the most likely conserved function of TssA, they should state it more strongly in the discussion. They should also provide a more detailed model of how they believe TssA works. I find it hard to imagine how it is working. Is it sitting on the end of the sheath and somehow sliding sheath subunits underneath it? How is Hcp getting in? How is TssA simultaneously binding to the sheath, allowing it to polymerize, and staying bound to the end. The authors need to formulate a model with a figure to try and interpret all the relevant data. I am not sure that "chaperone" is the correct description of TssA function. Chaperones are usually defined as proteins that promote assembly or folding of proteins, interacting transiently, but are not maintained as part of the complex. TssA is maintained as a component of the complex.

With respect to TagA function, this protein might be similar to a phage tail terminator, given that these proteins also bind to the end of the polymerizing tail and stop polymerization. It could be informative to mention this similarity as this may provide some insight into how TagA is working. The authors should also try to explain what the common domain in TssA and TagA might be doing (binding the sheath?). Have mutations ever been tested that only affect one domain of one of these proteins, or TssA. Does TagA displace TssA at the distal end of the sheath or do they both bind at once?

While this is an interesting paper, I am not sure if the impact is great enough for publication in EMBO J. In some regard the data seem to confirm previous results. The authors need to more strongly explain how their data markedly advances our knowledge of these proteins. In addition, two papers have come out recently pertaining to TagA and TssA (Santin et al., Nature Micro; Dix et al., Nature Comm). I realize that these papers may have emerged after submission of this paper, but the authors will now have to compare their results to what has been found in these papers. These papers may have an impact on the suitability of this manuscript for publication in EMBO J as some of their conclusions overlap with conclusions made here. The authors must clearly address this issue.

Referee #2:

The MS "Roles of TssA and TagA in the assembly and function of the Type VI secretion system" describes the function of the TssA and TagA proteins in *Vibrio Cholera* and *Pseudomonas aeruginosa*. The authors analyzed the location and function of the TssA proteins in VC and PAO1, developed a novel imaging tool that allows detection of individual proteins in dynamic assemblies and used it to evaluate the stoichiometry of TssA in the T6SS, and analyzed the location and function of the TagA protein in VC. This is a massive amount of excellent work. It partly overlaps with a paper describing TagA in *E.coli* that has just come out in Nature Microbiology. One of the main differences of in the findings of the two papers is that the phenotype of a TagA knockout in VC is virtually identical to that of the WT, whereas deletion of TagA in *E.coli* results in 5 times less efficient killing.

In my opinion, this is the most comprehensive functional analysis of TssA and TagA proteins published so far. Unlike the above-mentioned Nat Micro paper, this MS describes the complexity and idiosyncrasies of TssA and TagA proteins fully, without attempting to present a clearcut story because the behavior and function of these proteins are actually very complex. For example, this MS shows that over expression of TagA strongly inhibits the assembly of T6SS complexes. The Nat Micro paper does not report such an experiment in the *E.coli* system - an awful omission for a paper published in such a high profile journal. I find it hard to believe that such an experiment was not

attempted, but its outcome does not fully agree with the clearcut story presented in the Nat Micro paper.

However, for the same reason this MS being a comprehensive study, it is not well structured. I strongly believe that this MS represents a very valuable and very important contribution to the field, but it needs to be significantly modified to be fully appreciated by its intended audience.

Colloquial words like "seem", "appear", "tend" should not be avoided unless absolutely needed. The sentences containing these words have to be rewritten in a scientific style.

I do not think that the classification of T6SS into types or classes and subclasses is relevant for this study. This information can be deleted to simplify the nomenclature. In fact, better names and simpler nomenclature for TssA/TagA proteins is needed. The authors are encouraged to introduce a new nomenclature. A good, sensible nomenclature leads to clarity and it will be adopted by the entire field (and the paper well cited as a result!).

The description of function of TssA proteins in PAO1 should be combined with that of VC proteins. There is more work done on the VC proteins, so one way to approach this modification is to describe the similarity of function of all TssAs and then continue with more detailed description of the VC system.

Line 72. I did not know what a "hexaflexagon" was before I looked it up on Wikipedia. It says that a hexaflexagon is a folded piece of paper. I do not think a protein structure can or should resemble a hexaflexagon.

Line 160-162. The logics escapes me. The 5-fold decrease in the number of structures per cell agrees with the 5-fold decrease in activity in the EC system (the Nat Micro paper). On the other hand, the tagA-minus mutant has a WT-like phenotype in VC. I do not understand the 5-fold decrease (and, by the way, it is not simply ONLY five-fold - it is almost ONE LOG!) and 2-fold decrease in the contraction rate. So, shouldn't we see $5 \times 2 = 10$ times less activity? Please explain.

Line 190. Please spell out the definition of the HA-tag before using this acronym.

Line 200-201. I do not understand the logics of this sentence.

Line 206. Since TssA of *V. cholerae* is evolutionary very closely related to TssAEC. This sentence does not read well. Maybe very close in terms of the amino acid sequence (how close - what is the identity?) instead of "evolutionary closeness"?

Line 214. Is it possible that TssA does not dissociate but bleaches out or photo-converts?

Line 235. Photobleaching or photo conversion again?

Line 262. This should be the last subsection in the MS. The PA stuff should come before it - combined with the VC data.

It is unclear how the intensity of the spots was measured and plotted in Fig. 6c. Are we looking at the max intensity of a single brightest pixel or at an average intensity of a group of pixels? If yes, how many pixels are in a single group? How was the size of the group chosen?

Line 369. "limitation of Hcp protein availability"

Line 372. "partial cell killing" - The prey cells are not alive, but not dead, so are they undead?

Referee #3:

In this work, Schneider et al set out to characterize the roles of two ImpA-containing proteins of the T6SS, TssA and TagA in different bacteria. They show that TssA and TagA play different roles in

T6SS biogenesis, using very nice fluorescence microscopy analyses of T6SS dynamics in bacteria. The data appear solid and support the conclusions. The manuscript is well written and presented. My only concern is with the novelty of the work. While the current work does present some new aspects regarding these ImpA-containing proteins, two recent papers discuss these proteins and their role in the T6SS biogenesis and reach similar conclusions: 1) differences in ImpA-containing proteins by Dix et al, Nature Communications, 2018; 2) TagA stops and holds the T6SS sheath by Santin et al. Nature Microbiology, 2018. I assume they were not mentioned or discussed since they were probably published while this work was being written, but I think that the current work and its findings should be discussed and evaluated in light of the above mentioned papers.

1st Revision - authors' response

29th Apr 2019

We would like to thank the reviewers for their careful reading and assessment of our manuscript. We have addressed all issues that were raised and overall strengthened our manuscript by including new data as well as by rewriting and reorganizing the text. We now describe localization dynamics and the roles of three different classes of ImpA_N domain containing proteins. To the revised manuscript, we added description of localization of *P. aeruginosa* TssA1, which we show localizes to the baseplate during sheath assembly. Overall, we now show that proteins from one class of ImpA_N domain containing proteins localize to the distal end of sheaths during their assembly, the proteins from the second class localize to the membrane and outcompete the proteins from the first class and the third class of these proteins localize to the sheath assembly initiation site. We have reorganized and streamlined the introduction, results, figures and discussion to describe these three different protein classes based on their function as well as discussed the recently published papers. We believe that these changes make it easier for the readers to appreciate and follow the presented data. Overall, we are convinced that this comprehensive study will help to direct future research of the diverse ImpA_N domain containing proteins. The detailed responses to the issues raised by the reviewers are below in blue.

Referee #1:

This manuscript describes the functions of TssA and TagA proteins in the type VI secretion system (T6SS). The authors perform thorough in vivo fluorescence experiments that clearly demonstrate functions of TssA and TagA. They also provide data showing which T6SS proteins are bound by TssA and TagA.

This is an interesting paper with well produced data. However, the paper is confusing in parts. In the bioinformatic figure, there are full strain names included and it is hard to connect those names with the proteins that are being discussed in the paper. The authors say that certain proteins are closely related, but it is impossible to know what this means. What are the pairwise percent identities between *Vibrio* and *E. coli*? Are all domains conserved? What is the percent identity with the *Pseudomonas* proteins. In the section discussing these proteins, Uniprot accession numbers are confused with Pfam family names (e.g. line 128). I also found it difficult to understand what was being shown in the kymograph parts of Fig. 4 and Fig. 5. The authors need to explain this as not all readers are familiar with this type of figure.

We have simplified the description of the three classes of ImpA_N domain containing proteins. We removed the phylogenetic analysis and instead focused on classifying the various proteins based on their domain composition. In the Supplementary figure 1 we

now depict simple scheme of domain organization and show examples of proteins for each class.

We amended the text to better describe the kymographs before their first use (see the second paragraph of the second section of the results).

The authors use a "very sensitive" assay for activity of the Vibrio T6SS. The authors should discuss whether this is too sensitive an assay. I wonder how mutations of genes like *tssE*, which are absolutely conserved, could not result in full loss of activity (e.g. knocking out the homologous phage gene completely eliminates tail production). Is *E. coli* too easy to kill? Should activity also be tested against a more challenging enemy (e.g. amoeba).

We agree with the reviewer that the fact that *tssE* negative strain kills *E. coli* is surprising. However, we have to point out that this is consistent with the fact that such strain assembles sheaths, albeit at a highly reduced rate. Here we use *tssE* mutant background to assess the role of TagA protein in T6SS function. Since *tagA* negative strain kills *E. coli* almost as well as the parental strain, we use the *tssE* mutation as a way to significantly decrease the overall T6SS activity to reveal the role of TagA under conditions when sheath assembly rate is limited. We did this because the laboratory conditions and the high T6SS assembly rate of the *V. cholerae* strain may not fully reflect all conditions under which T6SS is used in nature. *tssE* negative strain is therefore our proxy for low T6SS assembly rate conditions. The reviewer is right that using a different T6SS target would be another option how to address this issue, however, amoeba is not killed by the strain of *V. cholerae* that we used in our study as this strain has a frame shift mutation in the *vgrG1* gene, which is essential for virulence. We thus decided not to pursue the generation of the required strains and establishment of new T6SS activity assays and rather strengthen other aspects of this manuscript.

The conclusions concerning TssA function are not described very clearly. The authors mention results from other studies that suggest a variety of functions for TssA. If the authors have an opinion about what is the most likely conserved function of TssA, they should state it more strongly in the discussion. They should also provide a more detailed model of how they believe TssA works. I find it hard to imagine how it is working. Is it sitting on the end of the sheath and somehow sliding sheath subunits underneath it? How is Hcp getting in? How is TssA simultaneously binding to the sheath, allowing it to polymerize, and staying bound to the end. The authors need to formulate a model with a figure to try and interpret all the relevant data. I am not sure that "chaperone" is the correct description of TssA function. Chaperones are usually defined as proteins that promote assembly or folding of proteins, interacting transiently, but are not maintained as part of the complex. TssA is maintained as a component of the complex.

We have rewritten both results and discussion to try to better describe the phenotypes of different ImpA_N domain containing proteins. We absolutely agree with the reviewer that the whole T6SS field would benefit from better understanding of the molecular mechanisms behind the functions of these proteins. However, this appears to be challenging and a detailed model will require significant insights into structures of these proteins and their interaction partners. We attempted this in the case of TssA_{VC}, however, the new insights are rather limited. Here, we hoped to improve current research by providing a systematic study of these proteins using consistent set of approaches. We agree that more needs to be done as this and other studies raised new questions.

We believe that the TssA proteins that help to assemble sheaths are in many ways similar to chaperones. As we now point out in the discussion, the assembly of the sheath-tube polymer requires proper formation of the four beta-stranded domain of the sheath and this likely requires assistance from TssA. While we show that assembly of this “hand-shake” domain is likely possible without TssA-like proteins, we also show that TssA significantly increases the speed of the sheath-tube co-assembly. We speculate that formation of the hand-shake domain requires folding of small parts of sheath subunit (linkers) and thus TssA could be the chaperone involved in this process. Since the sheath subunits are added at the distal end of the assembling sheath, TssA is dynamically interacting with new subunits and thus is not a “stable” part of an assembly. We agree with the reviewer that it is not clear how exactly this is achieved. Answering this question and having a precise model for a function of these proteins will surely require significant future effort.

With respect to TagA function, this protein might be similar to a phage tail terminator, given that these proteins also bind to the end of the polymerizing tail and stop polymerization. It could be informative to mention this similarity as this may provide some insight into how TagA is working. The authors should also try to explain what the common domain in TssA and TagA might be doing (binding the sheath?). Have mutations ever been tested that only affect one domain of one of these proteins, or TssA. Does TagA displace TssA at the distal end of the sheath or do they both bind at once?

This is an interesting suggestion. We searched for possible homology of TssA-like and TagA-like proteins to the phage tail-terminators (both contractile and non-contractile), however, we failed to detect any homology. We used HHpred server and initiated the homology searches either with TssA/TagA-like proteins or phage proteins but found no overlap in the results.

Concerning the TagA/TssA coordination, we think that our data indeed support the model where TssA is displaced from sheath by TagA. This is now described in the updated discussion.

While this is an interesting paper, I am not sure if the impact is great enough for publication in EMBO J. In some regard the data seem to confirm previous results. The authors need to more strongly explain how their data markedly advances our knowledge of these proteins. In addition, two papers have come out recently pertaining to TagA and TssA (Santin et al., Nature Micro; Dix et al., Nature Comm). I realize that these papers may have emerged after submission of this paper, but the authors will now have to compare their results to what has been found in these papers. These papers may have an impact on the suitability of this manuscript for publication in EMBO J as some of their conclusions overlap with conclusions made here. The authors must clearly address this issue.

Indeed, our manuscript was submitted before these manuscripts were published. We have amended our text with a comparison and discussion of these new data. We agree that there are overlaps between these studies, however, we are convinced that our comprehensive study is of significant value to the community.

Referee #2:

The MS "Roles of TssA and TagA in the assembly and function of the Type VI secretion system" describes the function of the TssA and TagA proteins in *Vibrio Cholera* and *Pseudomonas aeruginosa*. The authors analyzed the location and function of the TssA proteins in VC and PAO1, developed a novel imaging tool that allows detection of individual proteins in dynamic assemblies and used it to evaluate the stoichiometry of TssA in the T6SS, and analyzed the location and function of the TagA protein in VC. This is a massive amount of excellent work. It partly overlaps with a paper describing TagA in *E.coli* that has just come out in *Nature Microbiology*. One of the main differences of in the findings of the two papers is that the phenotype of a TagA knockout in VC is virtually identical to that of the WT, whereas deletion of TagA in *E.coli* results in 5 times less efficient killing.

We thank the reviewer for nice comments.

In my opinion, this is the most comprehensive functional analysis of TssA and TagA proteins published so far. Unlike the above-mentioned *Nat Micro* paper, this MS describes the complexity and idiosyncrasies of TssA and TagA proteins fully, without attempting to present a clearcut story because the behavior and function of these proteins are actually very complex. For example, this MS shows that over expression of TagA strongly inhibits the assembly of T6SS complexes. The *Nat Micro* paper does not report such an experiment in the *E.coli* system - an awful omission for a paper published in such a high profile journal. I find it hard to believe that such an experiment was not attempted, but its outcome does not fully agree with the clearcut story presented in the *Nat Micro* paper.

However, for the same reason this MS being a comprehensive study, it is not well structured. I strongly believe that this MS represents a very valuable and very important contribution to the field, but it needs to be significantly modified to be fully appreciated by its intended audience.

Colloquial words like "seem", "appear", "tend" should not be avoided unless absolutely needed. The sentences containing these words have to be rewritten in a scientific style.

We appreciate these comments and agree with the reviewer that the text needed improvements. We have indeed streamlined all sections of manuscript. This is to help the readers to navigate through the complex phenotypes associated with three different classes of ImpA_N domain containing proteins. We welcome input on the rewritten manuscript.

I do not think that the classification of T6SS into types or classes and subclasses is relevant for this study. This information can be deleted to simplify the nomenclature. In fact, better names and simpler nomenclature for TssA/TagA proteins is needed. The authors are encouraged to introduce a new nomenclature. A good, sensible nomenclature leads to clarity and it will be adopted by the entire field (and the paper well cited as a result!).

We have simplified the description of the different classes of ImpA_N domain containing proteins and proposed new nomenclature in the discussion based on the phenotypes of the tested proteins.

The description of function of TssA proteins in PAO1 should be combined with that of VC proteins. There is more work done on the VC proteins, so one way to approach this modification is to describe the similarity of function of all TssAs and then continue with more detailed description of the VC system.

We agree with reviewer and we indeed now group the results and figures based on functions of the proteins rather than based on their origin.

Line 72. I did not know what a "hexaflexagon" was before I looked it up on Wikipedia. It says that a hexaflexagon is a folded piece of paper. I do not think a protein structure can or should resemble a hexaflexagon.

This is now rewritten.

Line 160-162. The logics escapes me. The 5-fold decrease in the number of structures per cell agrees with the 5-fold decrease in activity in the EC system (the Nat Micro paper). On the other hand, the tagA-minus mutant has a WT-like phenotype in VC. I do not understand the 5-fold decrease (and, by the way, it is not simply ONLY five-fold - it is almost ONE LOG!) and 2-fold decrease in the contraction rate. So, shouldn't we see $5 \times 2 = 10$ times less activity? Please explain.

We measured that there is indeed about 5x less sheaths assembled in the TagA mutant. However, what is special about *V. cholerae* is that the sheaths are stable as a fully extended structure for a significant amount of time. Therefore, we also quantified sheath contractions as those more closely represent the events relevant for the overall T6SS activity. When we did that, we only measured modest (2-fold) decrease in the overall number of contractions per cell per unit of time. This is due to the fact that the sheaths in tagA negative cells simply contract almost immediately after assembly. We have changed wording of this section to make this clearer.

Line 190. Please spell out the definition of the HA-tag before using this acronym.

The text is amended.

Line 200-201. I do not understand the logics of this sentence.

This section was rewritten and this sentence is no longer present.

Line 206. Since TssA of *V. cholerae* is evolutionary very closely related to TssAEC. This sentence does not read well. Maybe very close in terms of the amino acid sequence (how close - what is the identity?) instead of "evolutionary closeness"?

We have added a Supplementary table S2, where we list percent identity and percent similarity for the TssA proteins.

Line 214. Is it possible that TssA does not dissociate but bleaches out or photo-converts?

Line 235. Photobleaching or photo conversion again?

We believe that while photobleaching is in principle possible, we would expect to see gradual decrease in TssA spot intensity, rather than complete lack of signal from one frame to another. TssA forms a dodecamer and our detection limit seems to be under 6 molecules of mNG per spot, therefore, for a spot to decrease its intensity below the detection limit at least 6 molecules of mNG in the same complex would have to photobleach at the same time. In general, we do not observe a lot of photobleaching when using mNG fusions, therefore we think that lack of a signal in these cases indeed correspond to dissociation of TssA from the sheath.

Line 262. This should be the last subsection in the MS. The PA stuff should come before it - combined with the VC data.

It is unclear how the intensity of the spots was measured and plotted in Fig. 6c. Are we looking at the max intensity of a single brightest pixel or at an average intensity of a group of pixels? If yes, how many pixels are in a single group? How was the size of the group chosen?

We have rearranged the text as requested. We have updated the Material method section to explain clearly how we measured fluorescence intensity of spots. Briefly, we measured average signal intensity of about 5x5 pixels around the center of a spot and then subtracted an average signal of a whole cytosol of the cell in which we detected the spot. This was then compared to an intensity of randomly picked 5x5 pixels in the cell (also after background subtraction).

Line 369. "limitation of Hcp protein availability"

Line 372. "partial cell killing" - The prey cells are not alive, but not dead, so are they undead?

This is now fixed.

Referee #3:

In this work, Schneider et al set out to characterize the roles of two ImpA-containing proteins of the T6SS, TssA and TagA in different bacteria. They show that TssA and TagA play different roles in T6SS biogenesis, using very nice fluorescence microscopy analyses of T6SS dynamics in bacteria. The data appear solid and support the conclusions. The manuscript is well written and presented. My only concern is with the novelty of the work. While the current work does present some new aspects regarding these ImpA-containing proteins, two recent papers discuss these proteins and their role in the T6SS biogenesis and reach similar conclusions: 1) differences in ImpA-containing proteins by Dix et al, Nature Communications, 2018; 2) TagA stops and holds the T6SS sheath by Santin et al. Nature Microbiology, 2018. I assume they were not mentioned or discussed since they were probably published while this work was being written, but I think that the current work and its findings should be discussed and evaluated in light of the above mentioned papers.

As mentioned above, to strengthen our study, we have added additional data, rewrote and streamlined the manuscript and discussed the recent publications.

2nd Editorial Decision

24th May 2019

Thank you for submitting a revised version of your manuscript. The manuscript has now been seen by two of original referees, who find that their main concerns have been addressed. They do, however, point out remaining minor issues that should be addressed with textual and figure changes. Please also resolve the following editorial issues that have to be dealt with before I can extend formal acceptance of the manuscript.

 REFEREE REPORTS:

Referee #1:

This revised manuscript is much clearer and better organized than the original. The authors have effectively addressed my concerns.

Some minor issues:

In the structure of TssAvc, is nt1 domain disordered? This point is not made clearly.

From the results with three examples, the authors conclude that all classA TssA proteins likely perform the same function. This conclusion would be strengthened if the authors mentioned the % sequence identity among the 3 proteins in question. If they are extremely similar in sequence, then the generality of the conclusion is not as strong.

What is Δ retS?

Fig. 6-The authors could use more distinctive colours for the proteins of interest, especially in part B.

Referee #2:

The new version of the MS is an even more comprehensive analysis of TssA-like and TagA-like proteins. The delivery has improved compared to the previous version, although the new additions did not necessarily made the paper easier to understand. It is still difficult to keep track of all the TssA and TagA proteins that behave differently in different systems besides having the same name.

Listed below are small changes that address the wording but now the conclusions.

Name the three classes of the proteins in the abstract.

L. 36-37. I am not sure if the processes of contraction are part of biogenesis. I thought that biogenesis is either an assembly or development or emergence.

L. 64. Reference should cite a paper describing this phenomenon in T6SS(iv) (which I think are PVCs, not a T6SS).

L. 530-540. The fluorescent quantification could be explained a little better. Maybe a small diagram can be used? For example, I do not understand how the error was calculated. Was the noise assumed to be Gaussian or Poisson? Was the unevenness of background taken into account when the error was calculated or only the variation of the spot intensity?

L 221. ...The highly conserved WEP-motif is also present in TssA proteins of *A. hydrophila*

If this motif was absent in some species, it would have been special. Otherwise, it is unclear why this motif is mentioned here with respect to that particular species.

L. 237. Generated homology model of TssAVC Nt2 domain was fitted and refined into cryo-EM reconstruction using molecular dynamics flexible fitting (MDFF).

Should read:

A homology model of TssAVC Nt2 domain generated (how?) was fitted and refined into cryo-EM reconstruction using molecular dynamics flexible fitting (MDFF).

245 The Class B of TssA-like proteins ^[1]_{SEP} Remove "The"

L. 252. ... after full assembly and thus less fully...
... and thus fewer fully...

L. 397. While potentially not being strictly essential, they act as chaperones assisting sheath-tube copolymerization, which likely requires assistance in proper formation of the sheath hand-shake domain (Ge et al., 2015; Kudryashev et al., 2015; ^[1]_{SEP}FYI, elongation of sheath polymerization (not the initiation!) in T4 does not require chaperones. PMID: 533896

L. 449 possible that certain sheath-tube assemblies evolved to be less stable ^[1]_{SEP} Less prone to contraction

L. 461 contracted sheaths (Planamente et al., 2016). Due to partial homology to the phage gp6 protein, ^[1]_{SEP} There is no such thing as 'partial homology'. Homology is a binary concept. This reviewer finds the bioinformatic analysis of the Planamente et al., 2016 paper flawed. Both, bioinformatics and structural analysis showed that TssF and TssG are gp6 homologs. TssA1 of PAO1 might interact with the baseplate, but the two proteins are not homologs.

Figure 6 is unnecessary complex. The proteins discussed in the paper are little squiggles that are difficult to spot. The figure is actually dominated by the fine structure of the membrane (one million lipid molecules). 95% of the information in the figure is unnecessary details. In the color code legend, the colors are almost impossible to assign to objects in the main panels (and I am not color blind). Basically, simplify the figure so that the TagA and TssA proteins are the major players and the rest is very schematic - the membrane should not have any detail and the rest of the structure could lose 80% of its features and this will not impair the results described in this MS.

Author Point-by-Point response.

Referee #1:

This revised manuscript is much clearer and better organized than the original. The authors have effectively addressed my concerns.

Some minor issues:

In the structure of TssAvc, is nt1 domain disordered? This point is not made clearly.

X-ray structure of Nt2 domain, which we see on 2D classes and focused 3D map, helped us to model Nt2, but not Nt1 domain. Only linkers to this domain are partially visible.

Nt2 domain is highly flexible relative to the CTD but properly folded in final structure. Nt1 domain would be even more flexible as it is on periphery of the protein assembly. This is now mentioned in the text.

From the results with three examples, the authors conclude that all classA TssA proteins

likely perform the same function. This conclusion would be strengthened if the authors mentioned the % sequence identity among the 3 proteins in question. If they are extremely similar in sequence, then the generality of the conclusion is not as strong.

This is now mentioned in the text, lines 122-123, 136-137 and 243-244.

What is $\Delta retS$?

We added a short sentence addressing use of $\Delta retS$ mutation background (lines 346-347).

Fig. 6-The authors could use more distinctive colours for the proteins of interest, especially in part B.

Figure 6 has been simplified accordingly.

Referee #2:

The new version of the MS is an even more comprehensive analysis of TssA-like and TagA-like proteins. The delivery has improved compared to the previous version, although the new additions did not necessarily make the paper easier to understand. It is still difficult to keep track of all the TssA and TagA proteins that behave differently in different systems besides having the same name.

Listed below are small changes that address the wording but not the conclusions.

Name the three classes of the proteins in the abstract.

We could not find a good way to implement this change. We believe that further work on the function of these proteins may result in even better naming so we would prefer to only suggest new names in the discussion at this point.

L. 36-37. I am not sure if the processes of contraction are part of biogenesis. I thought that biogenesis is either an assembly or development or emergence.

We modified the text accordingly.

L. 64. Reference should cite a paper describing this phenomenon in T6SS(iv) which I think are PVCs, not a T6SS).

We think the reviewer referred to lines 57-60. Data of Böck *et al* suggest that *Amoebophilus* Afp-like gene cluster encodes a T6SS rather than an eCIS (extracellular Contractile Injection System). In contrast to PVCs, ClpV is absent in TSS^{iv} and we added a second reference to Böck *et al* in line 61.

L. 530-540. The fluorescent quantification could be explained a little better. Maybe a small diagram can be used? For example, I do not understand how the error was calculated. Was the noise assumed to be Gaussian or Poisson? Was the unevenness of background taken into account when the error was calculated or only the variation of the spot intensity?

A diagram was added in Fig EV2C.

L. 221. ...The highly conserved WEP-motif is also present in TssA proteins of *A. hydrophila*. If this motif was absent in some species, it would have been special. Otherwise, it is unclear why this motif is mentioned here with respect to that particular species.

The text was changed accordingly (lines 219-220).

L. 237. Generated homology model of TssAVC Nt2 domain was fitted and refined into cryo-EM reconstruction using molecular dynamics flexible fitting (MDFF).

Should read:

A homology model of TssAVC Nt2 domain generated (how?) was fitted and refined into cryo-EM reconstruction using molecular dynamics flexible fitting (MDFF).

Changed the text accordingly. Process of model generation is described in material and methods corresponding to this section (lines 639-642).

245 The Class B of TssA-like proteins ^[1]_[SEP]Remove "The"

We removed this article.

L. 252. ... after full assembly and thus less fully...
... and thus fewer fully...

Text was amended accordingly.

L. 397. While potentially not being strictly essential, they act as chaperones assisting sheath-tube copolymerization, which likely requires assistance in proper formation of the sheath hand-shake domain (Ge et al., 2015; Kudryashev et al., 2015; ^[1]_[SEP]FYI, elongation of sheath polymerization (not the initiation!) in T4 does not require chaperones. PMID: 533896

We included this information in lines 394-395.

L. 449 possible that certain sheath-tube assemblies evolved to be less stable ^[1]_[SEP]Less prone to contraction

We changed the text as suggested.

L. 461 contracted sheaths (Planamente et al., 2016). Due to partial homology to the phage gp6 protein, ^[1]_[SEP]There is no such thing as 'partial homology'. Homology is a binary concept. This reviewer finds the bioinformatic analysis of the Planamente et al., 2016 paper flawed. Both, bioinformatics and structural analysis showed that TssF and TssG are gp6 homologs. TssA1 of PAO1 might interact with the baseplate, but the two proteins are not homologs.

We modified the text and deleted this statement (line 459).

Figure 6 is unnecessary complex. The proteins discussed in the paper are little squiggles that are difficult to spot. The figure is actually dominated by the fine structure of the membrane (one million lipid molecules). 95% of the information in the figure is unnecessary details. In the color code legend, the colors are almost impossible to assign to objects in the main panels (and I am not color blind). Basically, simplify the figure so that the TagA and TssA proteins are the major players and the rest is very schematic - the membrane should not have any detail

and the rest of the structure could lose 80% of its features and this will not impair the results described in this MS.

Figure 6 has been simplified accordingly.

Thank you very much for incorporating the final changes into the revised manuscript. I am now happy to inform you that your manuscript has been accepted for publication in the EMBO Journal. Congratulations!

USEFUL LINKS FOR COMPLETING THIS FORM

<http://www.antibodypedia.com>
<http://1degreebio.org>
<http://www.equator-network.org/reporting-guidelines/improving-bioscience-research-repor>

<http://grants.nih.gov/grants/olaw/olaw.htm>
<http://www.mrc.ac.uk/Ourresearch/Ethicsresearchguidance/Useofanimals/index.htm>
<http://ClinicalTrials.gov>
<http://www.consort-statement.org>
<http://www.consort-statement.org/checklists/view/32-consort/66-title>

<http://www.equator-network.org/reporting-guidelines/reporting-recommendations-for-tum>

<http://datadryad.org>

<http://figshare.com>

<http://www.ncbi.nlm.nih.gov/gap>

<http://www.ebi.ac.uk/ega>

<http://biomodels.net/>

<http://biomodels.net/miriam/>
<http://jil.biochem.sun.ac.za>
http://oba.od.nih.gov/biosecurity/biosecurity_documents.html
<http://www.selectagents.gov/>

Corresponding Author Name: Marek Basler
 Journal Submitted to: EMBO Journal
 Manuscript Number: EMBOJ-2018-100825R